# Towards Robust Real-World Multivariate Time Series Forecasting: A Unified Framework for Dependency, Asynchrony, and Missingness

**Jinkwan Jang**[*]   **Hyungjin Park**[*]   **Jinmyeong Choi**   **Taesup Kim**[†]
Graduate School of Data Science
Seoul National University

## Abstract

Real-world time series data are inherently multivariate, often exhibiting complex inter-channel dependencies. Each channel is typically sampled at its own period and is prone to missing values due to various practical and operational constraints. These characteristics pose three fundamental challenges involving channel dependency, sampling asynchrony, and missingness, all of which must be addressed simultaneously to enable robust and reliable forecasting in practical settings. However, existing architectures typically address only parts of these challenges in isolation and still rely on simplifying assumptions, leaving unresolved the combined challenges of asynchronous channel sampling, test-time missing blocks, and intricate inter-channel dependencies. To bridge this gap, we propose ChannelToken-Former, a Transformer-based forecasting framework with a flexible architecture designed to explicitly capture cross-channel interactions, accommodate channel-wise asynchronous sampling, and effectively handle missing values. Extensive experiments on public benchmark datasets reflecting practical settings, along with one private real-world industrial dataset, demonstrate the superior robustness and accuracy of ChannelTokenFormer under challenging real-world conditions.

## 1 Introduction

Accurate time series forecasting is critical in domains such as industrial monitoring (Jean-Pierre et al., 2024; Zhao et al., 2024b), energy systems (Yao et al., 2025; Nascimento et al., 2023; Hu, 2024), and healthcare (He & Chiang, 2025; Tang et al., 2024), where predictive insights directly influence operational safety, resource efficiency, and long-term outcomes. Yet, real-world time series forecasting faces many practical challenges, including irregular sampling, concept drift within channels, and uncertainty quantification. Among these, complex interdependencies among channels, channel-wise asynchrony, and block-wise missingness are often oversimplified by current modeling approaches, which typically assume channel-independent, regularly sampled, and complete time series. In this work, we explicitly address these three key challenges within a unified framework.

One key challenge lies in **the complex interdependencies among channels**. Signals from different sensors or subsystems are rarely independent; their interactions often encode latent correlative dynamics crucial for accurate forecasting. Some studies have directly adopted channel-dependent architectures (Zhang & Yan, 2023; Wang et al., 2024c; Liu et al., 2024c), exploring inter-channel dependencies under specific structural assumptions. As an alternative, many models employ a channel-independent design (Nie et al., 2023; Wang et al., 2024b), which has shown competitive robustness to distributional drift (Han et al., 2024b). Nevertheless, carefully designed channel-dependent approaches (Chen et al., 2024a; Lee et al., 2024) can still exploit interdependencies to provide richer predictive signals and deliver performance gains beyond channel-independent baselines.

---

[*]Equal contribution.   [†]Corresponding author.

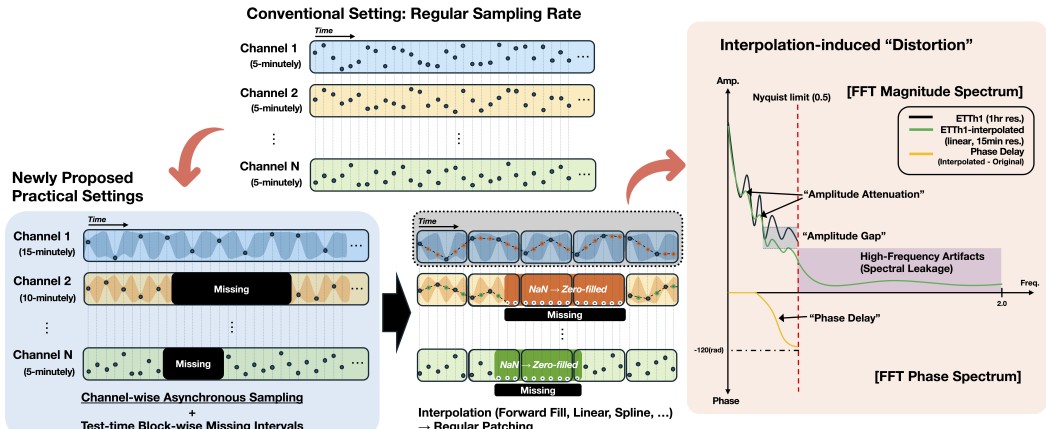

Figure 1: Our proposed practical conditions highlight the simultaneous presence of channel-wise asynchronous sampling, block-wise missingness at test time, and inter-channel dependencies. Interpolation over coarsely sampled regions leads to signal distortion. See Appendix B for more details.

Another difficulty arises from the heterogeneity of data sources (Reiss et al., 2019; Filho et al., 2024; Dong et al., 2025; Ying et al., 2025; Agency, 2025). Time series signals commonly originate from diverse sensors, such as those tracking temperature, pressure, actuator positions, or biological signals. Due to differing physical properties and application contexts, these signals are frequently sampled at varying temporal resolutions, often leading to **channel-wise (multi-source) asynchronous sampling** in practice. Nonetheless, most existing approaches (Wang et al., 2024c; Chen et al., 2025) assume idealized input conditions: fully observed sequences sampled at identical intervals and aligned timestamps across channels. Such assumptions ignore variations in sampling periods and sequence lengths, complicating both model design and data preprocessing in real-world settings. Beyond channel-independent strategies, the challenge of channel-wise asynchronous sampling remains underexplored.

As with interdependencies and sampling variability, missing data represents another key challenge in time series modeling. In practice, discrete missing values are already present in standard benchmarks such as ETTm, and modern neural time-series models can exhibit a certain degree of implicit robustness to such patterns (see Appendix C.6). However, real-world signals often contain **long contiguous intervals of missing observations**, often arising from maintenance issues, sensor malfunctions, or communication failures (You et al., 2025; Nejad et al., 2024). In such cases, naive interpolation, which is commonly applied during preprocessing, can be unreliable or misleading, particularly for dynamic signals. More robust approaches can instead benefit from leveraging cross-channel correlations to infer missing dynamics, rather than relying solely on extrapolation from partially observed inputs.

These three core challenges highlight the need for forecasting models that remain robust under realistic conditions, as illustrated in Figure 1. In particular, models should be capable of: (i) leveraging cross-channel dependencies to capture complex interactions and maintain structural consistency; (ii) handling asynchronous, variable-length inputs across channels without strict alignment or resampling; and (iii) addressing block-wise missing intervals during test-time inference by directly exploiting information from other channels, rather than relying on imputation methods.

**While these requirements are clear, prior approaches have only provided partial solutions.** Channel-dependent models capture inter-channel dependencies but overlook asynchrony and missingness; channel-independent strategies handle misalignment but lose cross-channel structure; and specialized approaches for missing values do not account for asynchrony, while irregular time series methods focus on within-channel non-uniformity under sparse settings rather than cross-channel heterogeneity in sampling.

To overcome these limitations, we propose ChannelTokenFormer, *a unified Transformer-based forecasting framework designed to tackle all three challenges in a unified and simultaneous manner: channel-wise asynchronous sampling, test-time missing blocks, and cross-channel dependencies.* The key is to revisit *channel tokens* under realistic multivariate conditions, viewing them as compact

representations that aggregate local temporal information within each channel while also capturing cross-channel dependencies. Although this idea resembles the channel-level summary tokens in iTransformer (Liu et al., 2024c) and TimeXer (Wang et al., 2024e), our framework distinguishes itself by introducing a mask-guided attention strategy that enables a unified treatment of intra- and cross-channel interactions. In this way, the tokens serve as global attention anchors. Predictions are made from compressed channel-wise representations that account for heterogeneous sampling periods and masked tokens from missing blocks, regardless of the number of local tokens. This design preserves natural channel resolutions, remains robust to partially observed inputs, and adaptively captures cross-channel dependencies.

## 2 RELATED WORK

**Channel-Dependent (CD) Strategies for Multivariate Time Series**  CD strategies jointly model all channels to capture inter-channel dependencies, and has been widely adopted across various neural architectures, including TCN/CNNs (Luo & Wang, 2024; Wu et al., 2023), MLPs (Han et al., 2024a; Huang et al., 2024), Transformers (Cheng et al., 2024; Liu et al., 2024b; Shu & Lampos, 2024; Wang et al., 2024e;c; Zhang & Yan, 2023; Liu et al., 2024c; Yang et al., 2024; Yu et al., 2023), and GNNs (Huang et al., 2023; Yi et al., 2023a; Cao et al., 2020; Liu et al., 2022). Some recent models (Chen et al., 2024a; Qiu et al., 2025) refine CD strategies by learning sparse or clustered inter-channel structures. Moreover, TimeFilter (Hu et al., 2025) constructs a spatial–temporal graph over patches and filters out spurious dependencies in a patch-specific manner. Generative approaches such as COSCI-GAN (Seyfi et al., 2022) also employ CD principles, emphasizing cross-channel coherence in the context of sequence generation. These approaches are particularly effective when channels are correlated, since sparsely observed or low-resolution signals can benefit from denser ones through cross-channel interactions. In practice, however, most CD strategies only partially address real-world challenges: they typically assume aligned sampling across channels, are not designed to handle block-wise missing intervals and can be sensitive to noisy inter-dependencies. In contrast, our method utilizes channel tokens that explicitly encode cross-channel relationships while remaining robust under asynchronous sampling and block-wise missingness.

**Irregularity of Multivariate Time Series (Multi-Source Asynchrony)**  Irregularly sampled multivariate time series are characterized by channels observed at non-uniform and often unpredictable time points, resulting in irregular gaps within individual channels. A number of recent works (Che et al., 2018; Shukla & Marlin, 2021; Li et al., 2023a; Chen et al., 2023; Zhang et al., 2023; 2024; Mercatali et al., 2024; Liu et al., 2024a; 2025; Kim & Lee, 2024; Klötergens et al., 2025) address this challenge by developing models that operate directly on non-uniform time intervals, often using time encodings, interpolation-based alignment, attention mechanisms, or graph-based structures. Models such as Raindrop (Zhang et al., 2022b) and Hi-Patch (Luo et al., 2025) are tailored for highly sparse settings, focusing on point-wise inter-channel relations rather than capturing temporal patterns. As a result, they are less suited to structured multi-source asynchrony and do not align with our proposed practical setting, which unifies three core challenges in continuous, long-term forecasting. In contrast, we target a real-world setting, including fixed but distinct sampling periods across channels, as categorized in the recent Time-IMM dataset (Chang et al., 2025).

**Missing Value Handling for Time Series and Test-time Missing Intervals**  Models such as BRITS (Cao et al., 2018) and SAITS (Du et al., 2023) reconstruct missing values through imputation based on recurrent dynamics or self-attention, while general-purpose frameworks like Times-Net (Wu et al., 2023) and TimeMixer++ (Wang et al., 2024a) treat imputation as a downstream task. While these models can perform imputation, they typically rely on a separate forecasting module to complete the end-to-end pipeline, which increases complexity and may hinder practical deployment in real-world scenarios. BiTGraph (Chen et al., 2024b) and S4M (Jing et al., 2025) integrate missing-value handling into forecasting architectures, but still assume regularly sampled time series. Other studies such as TimeXer (Wang et al., 2024e) and TFT (Zhou, 2023) test robustness to missing inputs by simple replacements, offering no explicit mechanism for structured sparsity. SERT (Nejad et al., 2024) highlights the importance of handling block-wise missingness in real-world forecasting scenarios, reinforcing the need for explicit mechanisms beyond simple replacement. In contrast, our approach leverages channel tokens to minimize imputation-induced distortion and directly handle test-time missing intervals during inference, providing a robust solution for real-world settings.

## 3 PROBLEM FORMULATION

**Asynchronous Channel-wise Observations** We consider a multivariate time series forecasting task with $N$ channels, where each channel $i \in \{1, \ldots, N\}$ is sampled at a distinct fixed period. Let $\{s_i\}_{i=1}^{N}$ denote the *relative* sampling periods across channels, normalized such that $\min(s_1, \ldots, s_N) = 1$. Given an input length $L$, the number of *valid* sampled points available for channel $i$ becomes $L_i = \lfloor L/s_i \rfloor$. Similarly, for a prediction horizon of length $H$, the number of future points to be predicted for channel $i$ is $H_i = \lfloor H/s_i \rfloor$. This adjustment is made in the data-point domain, not the time domain, so that the actual time span covered by the prediction remains consistent across channels despite different temporal resolutions.

Our learning objective computes the forecasting error *only at valid (observable) positions* for each channel, which is standard in irregular forecasting settings (Luo et al., 2025; Zhang et al., 2024). We refer to this as *Channel-aggregated MSE (CMSE)*:

$$\mathcal{L}_{\text{total}} = \frac{1}{N} \sum_{i=1}^{N} \mathcal{L}^{(i)} = \frac{1}{N} \sum_{i=1}^{N} \sum_{j=1}^{H_i} \frac{1}{H_i} \left( y_j^{(i)} - \hat{y}_j^{(i)} \right)^2. \tag{1}$$

For evaluation, we adopt CMSE as the primary metric, along with CMAE, its MAE-based variant.

**Test-time Missing Intervals** In many real-world cases, observations exhibit persistent sparsity due to temporary failures, communication losses, or maintenance, leading to block-wise missing intervals at test time, as shown in Figure 1. This type of missingness is typically easy to detect, as most channels rarely show prolonged zero readings unless the device is disconnected or malfunctioning.

Conventional patch-based methods cannot handle variable numbers of patches per channel, and thus typically fill missing entries within each patch using zeros or mean values. In contrast, our model adopts a non-overlapping patch representation that supports variable patch lengths and counts across channels. This enables the removing fully unobserved patches from the input, rather than merely filling in missing values. This formulation supports forecasting under heterogeneous horizons from channel-wise asynchronous sampling, while also accommodating channel-specific missingness in the input at test time. The forecast horizon is assumed to be fully observed, since missing intervals in the future cannot be predicted in advance.

## 4 CHANNELTOKENFORMER

**Repurposing Channel Tokens for Realistic Settings** Pioneering studies have explored the use of auxiliary tokens, for example, query tokens for extracting forecasting-relevant information from sequences of varying lengths (Kim et al., 2024), and special tokens for mediating information exchange across heterogeneous entities (Zhao et al., 2024a). Channel-level summary tokens have also been introduced in iTransformer (Liu et al., 2024c) and TimeXer (Wang et al., 2024e). Building on these insights, we repurpose channel tokens for realistic multivariate forecasting conditions, where patches are inherently unbalanced across channels due to heterogeneous sampling, optionally their own dominant frequencies, and prolonged missing intervals. In our framework, Channel Tokens act as compact abstractions that summarize each channel's patches into stable channel-level embeddings, so that forecasting relies on these embeddings rather than on uneven patch sequences that would otherwise complicate the decoder input. This reframing allows the model to remain robust under practical conditions where patch imbalance poses a fundamental challenge.

**Channel-wise Frequency-based Dynamic Patching and Tokenization** Prior frequency-guided patching (e.g., Moirai (Woo et al., 2024), LightGTS (Wang et al., 2025)) in time series foundation models targeted variable-length segmentation for *univariate* streams with heterogeneous granularities. In contrast, we repurpose this frequency-guided granularity selection for a *channel-wise asynchronous* multivariate setting. The detailed procedure is described in Algorithm 1. This enables non-aligned per-channel patching while preserving efficiency by sharing projection layers across channels with the same patch length and introducing dedicated channel tokens. For each channel, we estimate a dominant period via FFT to determine the patch length, and use a sampling-aware fallback when no clear peak exists. We then split each channel into non-overlapping patches and

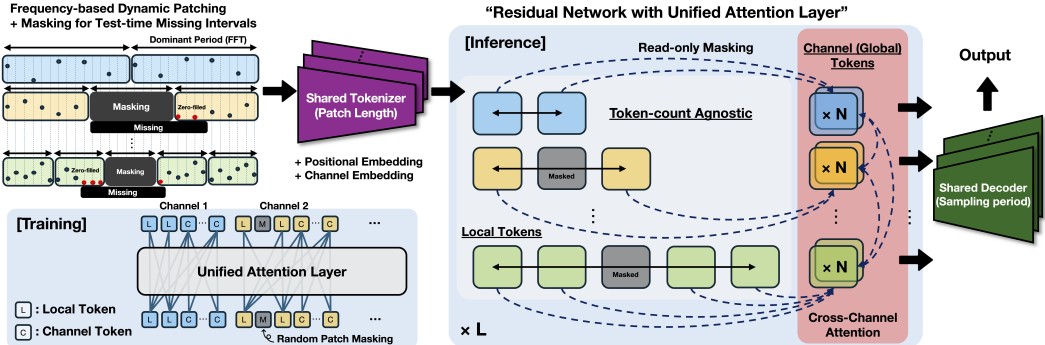

Figure 2: Overview of ChannelTokenFormer (CTF). All tokens across channels pass through a unified attention layer, where local and global information is aggregated into channel tokens. Only the channel tokens are decoded by decoders, each shared among channels with the same sampling period, to produce the final prediction.

map each patch to a local token. For each channel, we also initialize a small set of channel tokens. Each local token receives a fixed positional embedding $\mathbf{e}_{\text{pos}}$ and a learnable channel embedding $\mathbf{e}_{\text{ch}}^{(i)}$, while the channel token receives only $\mathbf{e}_{\text{ch}}^{(i)}$. Full details on thresholds, dataset-specific examples, and the fallback procedure are provided in Appendix A.2.

**Training-Time Proxying for Test-Time Missingness via Channel-wise Patch Masking**   Building on channel-wise dynamic patching, our framework supports variable input lengths via patch masking and explicitly targets block-wise missingness at test time. Unlike prior uses of random patch masking that primarily serve as a regularizer to curb overfitting, *we repurpose it as a training-time proxy for realistic test-time incompleteness*: inference-time inputs may be partially missing, so the model is trained to handle such missing blocks by design. We adopt random patch masking, as introduced in PatchDropout (Liu et al., 2023), applied to a subset of channel-wise patches at the input stage during training to simulate block-wise missingness at test time. As illustrated in Figure 2, if a missing segment spans an entire patch at test time, the corresponding local token is removed at the input stage, thereby being excluded from the attention computation. This prevents fully unobserved patches from contributing spurious signals, minimizing the risk of propagating invalid information. Despite temporal gaps induced by masking, temporal ordering and context are retained through fixed positional embeddings. This perspective shift improves resilience to test-time missing intervals and, as a side effect, acts as an implicit regularizer that mitigates overfitting in low-resource settings. This procedure is formalized in Algorithm 1.

**A Unified Mask-Guided Attention with Channel Tokens for Real-World Challenges**   To jointly address three practical challenges in multivariate time series, we design a *unified, mask-guided* self-attention mechanism that integrates local and global representations within a single attention operation. As illustrated in Figure 2, this design unifies both token types in one attention step. Each channel $i \in \{1, \ldots, N\}$ contributes two types of tokens: (i) local tokens $\mathbf{T}^{(i)} \in \mathbb{R}^{\mathcal{T}^{(i)} \times d}$ representing patch-level embeddings, and (ii) global channel tokens $\mathbf{C}^{(i)} \in \mathbb{R}^{m \times d}$ summarizing high-level contextual information, where $\mathcal{T}^{(i)}$ and $C$ denote the number of local and channel tokens, respectively. The full token sequence is constructed as:

$$\mathbf{X} = [\mathbf{T}^{(1)}; \mathbf{C}^{(1)}; \ldots; \mathbf{T}^{(N)}; \mathbf{C}^{(N)}] \in \mathbb{R}^{\mathcal{T} \times d}, \text{ where } \mathcal{T} = \sum_{i=1}^{N}(\mathcal{T}^{(i)} + m). \quad (2)$$

This unified sequence is passed through a masked multi-head self-attention layer with residual connection:

$$\mathbf{X}_{\text{out}} = \mathbf{X} + \text{Attention}(Q, K, V) = \mathbf{X} + \text{softmax}\left(\frac{QK^{\top}}{\sqrt{d}} + \mathbf{M}\right) V. \quad (3)$$

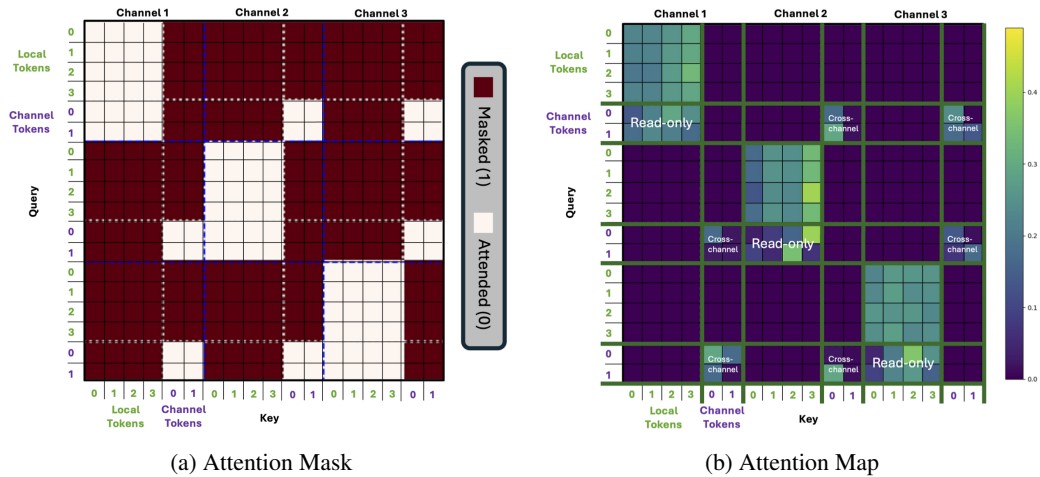

(a) Attention Mask
(b) Attention Map

Figure 3: Our unified attention masking strategy. Local tokens perform intra-temporal attention within the same channel. Channel tokens aggregate local and cross-channel information, but are not visible to local tokens and do not attend to themselves. Optionally, attention among channel tokens from the same channel can be masked to encourage inter-channel interaction and reduce redundancy.

Here, $\mathbf{M} \in \mathbb{R}^{\mathcal{T} \times \mathcal{T}}$ encodes structural constraints on how tokens can attend to one another, reflecting their types, channels, and masking. Our tailored masking scheme constructs $\mathbf{M}$ as illustrated in Figure 3: (1) Local tokens attend only to other local tokens within the same channel, enabling intra-temporal modeling. (2) Channel tokens attend to their own local tokens and to other channels' tokens, but are not accessible to local tokens due to their *read-only* role in the attention mechanism. (3) Channel tokens do not attend to themselves; for each query $\mathbf{C}^{(i)}$, the key $\mathbf{C}^{(i)}$ is excluded to avoid self-reinforcement and to encourage informative cross-channel interaction. Our unified masking strategy, inspired by read-only prompt attention (Lee et al., 2023) and attention control techniques (Kim et al., 2025), is implemented without modifying the standard Transformer architecture. Stacked masked attention blocks with residual connections and layer normalization enable deeper modeling of temporal and cross-channel dependencies. Although channel-wise tokenization introduces additional parameters and our unified attention remains quadratic in the total number of tokens, the proposed masking scheme focuses attention on the most relevant token interactions, and empirically does not lead to prohibitive inference latency for realistic channel counts and sequence lengths (see Appendix C.3).

## 5 EXPERIMENTS

To assess the effectiveness of ChannelTokenFormer (CTF) under our proposed problem settings, we conduct extensive experiments on four widely used multivariate time series benchmarks adapted to practical conditions, and on two real-world datasets from air monitoring and the LNG cargo handling system. Additional results under conventional settings are provided in Appendix D.

**Datasets** We evaluate forecasting performance under channel-wise asynchronous sampling using four multivariate datasets. **ETT1-practical and ETT2-practical (ETT1 & ETT2)** modify ETTm1 and ETTm2 (Zhou et al., 2021) by resampling channels to domain-specific temporal resolutions (e.g., 1-hour for load, 15-minute for temperature). **Weather-practical (Weather)**, adapted from the Weather benchmark (Wu et al., 2021), assigns heterogeneous sampling periods to channels based on distinct physical characteristics. **Monash-SolarWindPower-practical (SolarWind)** combines solar and wind power series from the Monash Forecasting Archive (Godahewa et al., 2021), with variables resampled to 20-minute and 5-minute intervals, respectively. **EPA-Air (EPA)**, adapted from the U.S. Environmental Protection Agency's air quality monitoring data (Chang et al., 2025), collects measurements from several regions with heterogeneous sampling periods across channels (e.g., 1-hour for temperature, 8-hour for $PM_{2.5}$, among others). We choose four regions, Maricopa, Richmond, LA and Hillsborough. **LNG Cargo Handling System (CHS)** is a real-world industrial dataset collected from an LNG carrier, consisting of sensor channels related to cargo operations,

ship navigation, and surrounding weather conditions. To assess robustness under block-wise test-time missing intervals, we additionally conduct experiments on SolarWind by varying the missing ratio. Detailed dataset specifications are provided in Appendix A.1.

**Baselines** We compare CTF against representative state-of-the-art models across architectures and modeling strategies that target inter-channel dependencies, irregular sampling, or missingness. Transformer baselines include TimeXer (Wang et al., 2024e), iTransformer (Liu et al., 2024c), and PatchTST (Nie et al., 2023). We also cover CNNs, GNNs, and MLPs, including TimesNet (Wu et al., 2023), CrossGNN (Huang et al., 2023), TimeMixer++ (Wang et al., 2024a), and DLinear (Zeng et al., 2023). We further include strong recent channel-dependent methods, DUET (Qiu et al., 2025) and TimeFilter (Hu et al., 2025); irregular-sampling methods such as t-PatchGNN (Zhang et al., 2024) and Hi-Patch (Luo et al., 2025); and a missingness-robust approach, BiTGraph (Chen et al., 2024b).

**Setup and Implementation Details** Baseline models for regular time series assume uniformly sampled and fully observed inputs. To satisfy this requirement, we employ linear interpolation to recover values unobserved at timestamps not covered by channel-specific sampling periods. To isolate architectural effects from interpolation, we further implement interpolation-free variants, including a version of TimeXer (Wang et al., 2024e) modified to handle temporally non-aligned cross-channel inputs. Implementation details are provided in Appendix C.4. In contrast, irregular or missing-aware baselines do not require interpolation, as they operate solely on observed values. To simulate block-wise missingness at test time, we randomly mask contiguous input regions. For fairness, regular baselines without explicit missing-value handling are also provided with linearly interpolated values for these masked regions, consistent with their training procedure. For each dataset, input and prediction lengths are selected to reflect the channel-wise temporal resolution. Details for each dataset are in Appendix A.2. Despite heterogeneous sampling periods across channels, it is practical to define input and prediction windows by a fixed time duration instead of requiring the same counts of sample points across channels. The actual number of input and output points per channel is determined by its sampling period, with shorter periods yielding more points over the same window.

## 5.1 MAIN RESULTS

**Case 1: Channel-wise Asynchronous Forecasting** We evaluate forecasting performance using datasets that reflect realistic sampling heterogeneity across channels. Unlike standard baselines that rely on interpolation, our approach preserves the original sampling structure and operates directly on observed values. When we average CMSE and CMAE over all prediction lengths (Table 1), CTF delivers strong overall performance, achieving the best or second-best results across datasets. This underscores the benefit of avoiding interpolation when channels are temporally misaligned. In particular, CTF handles asynchrony more effectively than irregular modeling on most datasets. Detailed results for each prediction length and dataset are provided in Appendix E. As CTF uses frequency-domain information for dynamic patching, we also compare it against frequency-based approaches and still consistently achieve superior performance (see Appendix C.5 for more details).

Table 1: Forecasting performance on the channel-wise asynchrony (Case 1).

| Approach | Channel-Dependent | | | | | | | Channel-Independent | | | Irregular modeling | |
|---|---|---|---|---|---|---|---|---|---|---|---|---|
| Model | CTF(ours) | TimeFilter | DUET | TimeXer | iTrans. | CrossGNN | TimesNet | TimeMixer++ | PatchTST | DLinear | Hi-Patch | tPatchGNN |
| Metric | CMSE CMAE | CMSE CMAE | CMSE CMAE | CMSE CMAE | CMSE CMAE | CMSE CMAE | CMSE CMAE | CMSE CMAE | CMSE CMAE | CMSE CMAE | CMSE CMAE | CMSE CMAE |
| ETT1 | **0.399** **0.410** | 0.412 0.418 | 0.424 0.425 | 0.422 0.424 | 0.435 0.431 | 0.428 0.416 | 0.452 0.444 | 0.433 0.432 | 0.411 0.416 | 0.425 0.420 | 0.448 0.454 | 0.465 0.470 |
| ETT2 | **0.377** **0.383** | 0.383 0.388 | 0.388 0.389 | 0.380 0.388 | 0.396 0.396 | 0.386 0.388 | 0.399 0.396 | 0.396 0.400 | 0.390 0.396 | 0.455 0.447 | 0.397 0.401 | 0.393 0.406 |
| SolarWind | **0.403** 0.452 | 0.404 **0.449** | 0.438 0.491 | 0.424 0.469 | 0.470 0.485 | 0.465 0.478 | 0.471 0.474 | 0.429 0.468 | 0.417 0.467 | 0.421 0.516 | 0.431 0.473 | 0.447 0.493 |
| Weather | 0.275 0.296 | **0.273** **0.293** | 0.309 0.322 | 0.300 0.310 | 0.313 0.323 | 0.285 0.315 | 0.314 0.322 | 0.276 0.296 | 0.275 0.331 | 0.287 0.331 | 0.301 0.320 | 0.312 0.324 |
| EPA | **0.776** 0.586 | 0.863 0.599 | 0.782 **0.579** | 0.886 0.611 | 0.882 0.611 | 0.979 0.665 | 0.937 0.643 | 0.931 0.637 | 0.854 0.597 | 1.047 0.713 | 0.808 0.629 | 0.801 0.628 |
| CHS | **0.285** 0.126 | 0.304 0.129 | 0.307 0.133 | 0.298 0.128 | 0.305 0.132 | 0.298 0.139 | 0.330 0.140 | 0.296 0.129 | 0.315 0.132 | 0.351 0.248 | 0.301 0.129 | 0.294 **0.125** |

**Case 2: Channel-wise Asynchronous Forecasting with Test-time Missing Blocks** We further evaluate all models under a more challenging setting, where test-time inputs include *block-wise missing intervals*, beyond discrete missing values, explicitly indicated to the model, on top of channel-wise asynchronous sampling. This configuration encompasses all three core challenges of interest.

In this case, all models receive inputs containing contiguous missing segments filled with zeros. But for a fair comparison, linear interpolation is applied to regular baselines to avoid underestimating their capability. Unlike other approaches, CTF can internally adjust input length by design, enabling the use of zero-patch masking to effectively handle missing regions. As shown in Table 2, our model consistently outperforms most baselines in average CMSE and CMAE for overall prediction lengths across different missing ratios, maintaining high performance even as the severity of missingness increases. This robustness stems from the use of random patch masking during training, which improves the model's ability to generalize to incomplete patterns. Unlike other methods, CTF avoids distortion from artificially zero-filled values and preserves signal fidelity under block-wise missing conditions. Detailed results across datasets are provided in Appendix E. The corresponding details are provided in Appendix A.2.

Table 2: Channel-wise Asynchronous Forecasting performance on the SolarWind dataset under block-wise test-time missingness with varying missing ratios $m$ (Case 2).

| Approach | Channel-Dependent | | | | | | | Channel-Independent | | | Irregular modeling | | Missing |
|---|---|---|---|---|---|---|---|---|---|---|---|---|---|
| Model | CTF(ours) | TimeFilter | DUET | TimeXer | iTrans. | CrossGNN | TimesNet | TimeMixer++ | PatchTST | DLinear | Hi-Patch | tPatchGNN | BiTGraph |
| Metric | CMSE CMAE | CMSE CMAE | CMSE CMAE | CMSE CMAE | CMSE CMAE | CMSE CMAE | CMSE CMAE | CMSE CMAE | CMSE CMAE | CMSE CMAE | CMSE CMAE | CMSE CMAE | CMSE CMAE |
| $m = 0.125$ | **0.409** **0.463** | 0.430 0.466 | 0.445 0.499 | 0.427 0.474 | 0.475 0.491 | 0.472 0.488 | 0.468 0.474 | 0.436 0.473 | 0.426 0.477 | 0.426 0.523 | 0.427 0.473 | 0.450 0.496 | 0.426 0.508 |
| $m = 0.250$ | **0.429** 0.482 | 0.444 **0.477** | 0.470 0.525 | 0.442 0.488 | 0.496 0.512 | 0.487 0.505 | 0.495 0.495 | 0.467 0.499 | 0.456 0.509 | 0.436 0.538 | 0.450 0.495 | 0.467 0.514 | 0.444 0.523 |
| $m = 0.375$ | **0.452** 0.507 | 0.471 **0.498** | 0.520 0.569 | 0.462 0.505 | 0.539 0.548 | 0.511 0.529 | 0.537 0.527 | 0.531 0.545 | 0.515 0.561 | 0.452 0.559 | 0.468 0.513 | 0.496 0.542 | 0.467 0.542 |
| $m = 0.500$ | **0.475** **0.533** | 0.522 0.533 | 0.599 0.629 | 0.514 0.543 | 0.606 0.598 | 0.553 0.562 | 0.595 0.564 | 0.633 0.613 | 0.593 0.616 | 0.478 0.587 | 0.500 0.542 | 0.522 0.571 | 0.495 0.562 |

## 5.2 ABLATION STUDY

To assess the effectiveness of our design in addressing the three key challenges, we conduct an ablation study where each component is selectively removed or replaced. For *Dependency*, we disable cross-channel attention. For *Asynchrony*, we remove the dynamic patching mechanism, forcing fixed patch boundaries that ignore dominant frequencies and channel-specific sampling periods. For *Missingness*, we eliminate patch masking during training and test time, exposing the model to zero-filled inputs instead. These experiments show that robust forecast-

Table 3: Ablation study under channel-wise asynchronous sampling with block-wise test-time missing (missing ratio = 0.375). Results are averaged across all prediction lengths on the SolarWind dataset.

| Ablation Setting | CMSE | CMAE |
|---|---|---|
| **Full model (CTF)** | **0.452** | **0.508** |
| w/o Channel Dependence (CI only) | 0.474 | 0.521 |
| w/o Dynamic patching | 0.494 | 0.536 |
| w/o Patch masking | 0.458 | 0.508 |

ing under practical conditions requires all three components, since removing any one degrades performance. Additional ablations are provided in Appendix C.1.

## 6 ANALYSIS

### 6.1 UNDERSTANDING THE OPTIMAL NUMBER OF CHANNEL TOKENS

The number of channel tokens was varied to analyze its impact on asynchronous forecasting. Results in Table 4 show that the optimal choice differs by dataset: two channel tokens work best for ETT1 and SolarWind, three for EPA, and one for Weather and CHS. These outcomes reflect how channels interact under asynchronous sampling. When channels move in a similar and highly correlated way, a single channel token captures the shared pattern and more tokens add redundancy. ETT1 and SolarWind, which exhibit two dominant groups of channels, benefit from two tokens. EPA shows stronger heterogeneity, and three tokens better capture its diverse structures. Overall, the optimal number of channel tokens depends on the degree of channel similarity and diversity in each dataset.

### 6.2 ROBUSTNESS TO INPUT LENGTH VARIABILITY

Random patch masking is primarily designed to make the model robust to incomplete inputs caused by block-wise missing intervals at test time. An additional benefit of this design is robustness to input length variability, which remains a limitation for conventional forecasting models. Most existing approaches train and evaluate separate models for each target input length, and even models that technically accept variable lengths often degrade in performance when the sequence length changes.

Table 4: Effect of varying the number of channel tokens {1, 2, 3} on asynchronous forecasting performance. Results are reported in CMSE, averaged over all prediction lengths.

| # Channel Tokens | Dataset | | | | | |
|---|---|---|---|---|---|---|
| | ETT1 | ETT2 | SolarWind | Weather | EPA | CHS |
| 1 | 0.399 | 0.377 | 0.404 | **0.274** | 0.775 | **0.282** |
| 2 | **0.398** | 0.377 | **0.400** | 0.275 | 0.777 | 0.289 |
| 3 | 0.411 | **0.375** | 0.406 | 0.281 | **0.774** | 0.295 |

Table 5: Evaluation of robustness to varying input lengths during test time. Our CTF is trained with an input length of 576 using random patch masking, and tested with other shorter input lengths of 288, 360, and 432 on the SolarWind dataset.

| Test Input Length | 288 | 360 | 432 | 576 |
|---|---|---|---|---|
| **Avg. CMSE** | 0.429 | 0.461 | 0.448 | 0.409 |
| **Avg. CMAE** | 0.457 | 0.495 | 0.491 | 0.457 |

As shown in Table 5, CTF maintains stable performance across all test input lengths on the Solar-Wind dataset without requiring retraining. Since each channel has a different sampling period, we define the input length based on the channel with the smallest sampling interval, which corresponds to 576 time steps over 2 days. Even when this length is reduced to 288 (equivalent to 1 day), the CMSE increases by only 4.9%, demonstrating strong generalization to input length variation, which is another essential property for real-world forecasting under inconsistent input conditions.

## 6.3 FREQUENCY BIAS ANALYSIS IN PREDICTED TIME SERIES

We analyze how interpolation affects the frequency characteristics of predicted time series under practical conditions with channel-wise asynchronous sampling. To this end, we focus on TimeXer (Wang et al., 2024e), which is structurally similar to CTF and achieves nearly identical performance to ours in the conventional setting on the ETTm1 benchmark. This makes TimeXer a strong reference point for fair comparison. In the practical setting, however, TimeXer can only process asynchronous inputs by performing interpolation, since its original design cannot directly accommodate non-aligned sequences without architectural modifications. As a result, any performance gap observed here reflects interpolation-induced distortion rather than inherent modeling differences. We analyze *dominant frequency difference* and *amplitude spectrum RMSE* across frequency bands in the ETT1 dataset (Table 6). In ETT1 (Table 6), both models localize dominant cycles similarly, since the main periodicities (e.g., 8, 12, and 24 hours) lie in the low- to mid-frequency range where interpolation-induced phase delay is minimal. However, amplitude spectrum RMSE shows consistent gaps: as Figure 4 illustrates, both models exhibit some attenuation, but the effect is noticeably stronger for TimeXer, while CTF better preserves spectral energy. Thus, although TimeXer matches CTF in conventional settings, its reliance on interpolation in the practical setting introduces frequency bias that propagates into predictions. By avoiding interpolation, CTF mitigates this distortion and better retains trend and periodicity under asynchronous sampling.

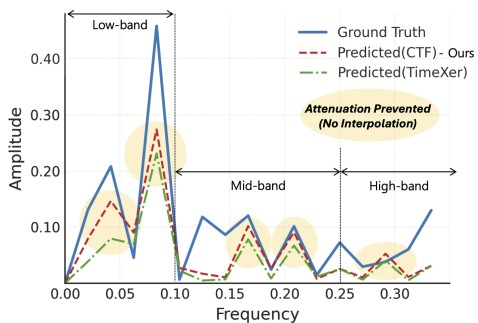

Figure 4: Frequency-domain comparison between CTF and TimeXer on a test sample from ETT1, showing that amplitude attenuation is prevented across all frequency bands.

Table 6: Frequency-domain comparison between CTF (interpolation-free) and TimeXer (interpolation-based) on the ETT1 for a prediction length of 192 (2-day window). Lower values indicate better spectral fidelity. RMSEs are computed over three frequency bands under 1-hour sampling ($f_s = 1$): *Low* (0.00–0.10), *Mid* (0.10–0.25), and *High* (0.25–0.50) cycles/hour, corresponding to periodicities longer than 10 hours, between 4 and 10 hours, and shorter than 4 hours, respectively.

| Model / Metric | Dominant Freq. Diff. | Low-band RMSE | Mid-band RMSE | High-band RMSE |
|---|---|---|---|---|
| CTF (Ours) | **0.0098** | **0.1877** | **0.0735** | **0.0457** |
| TimeXer | 0.0099 | 0.1901 | 0.0745 | 0.0473 |

## 7    CONCLUSION

A practical forecasting setting is introduced where three core real-world challenges occur simultaneously: channel-wise asynchronous sampling, block-wise missing intervals, and complex cross-channel dependencies. Instead of interpolation, our approach handles both missing segments and sampling gaps through masking and frequency-based dynamic patching. The unified mask-guided attention leverages channel tokens with read-only masking and cross-channel attention, consistently outperforming prior methods across real-world conditions. Our scalability analysis indicates that the masking-guided unified attention design remains computationally manageable for realistic channel counts and input lengths. However, ultra-high-dimensional regimes with thousands of channels (e.g., Traffic) will require additional mechanisms such as explicit channel sparsification or grouped channel representations, which we leave for future work. Beyond scaling, future work will also consider applying ChannelTokenFormer to broader domains, integrating multimodal signals, and improving interpretability of channel interactions, thereby expanding its utility in real-world forecasting.

## ACKNOWLEDGMENTS

This work was supported by the National Research Foundation of Korea (NRF) grant funded by the Korea government (MSIT) (No. RS-2024-00345809, Research on AI Robustness Against Distribution Shift in Real-World Scenarios; and No. RS-2023-00222663, Center for Optimizing Hyperscale AI Models and Platforms). We sincerely thank HD Korea Shipbuilding & Offshore Engineering Co., Ltd. (HD KSOE), particularly the Thermal Systems Research Lab, for granting access to proprietary data under a confidentiality agreement and for supporting the secure data-sharing process that enabled this research.

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

# A  IMPLEMENTATION DETAILS

## A.1  DATASET CONFIGURATION UNDER MULTI-SOURCE ASYNCHRONY

We conduct long-term forecasting experiments on four modified benchmark datasets adapted to our practical setting, as well as two real-world datasets. In this setting, each channel provides a different number of observations over the same temporal window, reflecting channel-specific sampling periods within the multivariate time series. This configuration more closely mirrors real-world conditions, where sensor deployment strategies and data acquisition policies vary across channels.

To further contextualize this setting, we note that widely used benchmarks already involve substantial preprocessing to obtain complete, regularly sampled grids. For example, the ETT datasets typically aggregate higher-frequency sensor streams for load-related channels into 15-minute or hourly averages. This procedure reduces the statistical complexity of the original processes and introduces its own form of synthetic regularity. Our *practical variants* do not create any new values; instead, they relax this enforced uniformity by selectively downsampling channels whose variability, assessed in the frequency domain, indicates sufficient oversampling. The resulting multi-rate sensing patterns more plausibly reflect real-world conditions while not materially altering the underlying temporal dynamics.

**ETT1-practical & ETT2-practical (ETT1 & ETT2)**  ETT1-practical and ETT2-practical are variants of the commonly used ETTm1 (Zhou et al., 2021) and ETTm2 datasets. Unlike their original versions, where all channels are sampled uniformly at 15-minute intervals, ETT1-practical and ETT2-practical introduce channel-specific sampling periods to reflect industrial environments, where sensors operate at different acquisition frequencies depending on their physical properties and monitoring requirements. Specifically, the channels are grouped into two categories: load-related channels and the oil-temperature (OT) channel. The load-related channels are downsampled to an hourly rate, representing slowly varying electrical load measurements, while the OT channel retains its original 15-minute resolution to capture high-frequency thermal dynamics.

**Monash-SolarWind-practical (SolarWind)**  Monash-SolarWind-practical is derived from Australian Energy Market Operator (AEMO) [1] and the Monash Forecasting Repository (Godahewa et al., 2021) and built upon high-frequency power generation data originally collected at 4-second intervals. In real-world energy monitoring systems, distinct sensors are used for solar and wind power, each exhibiting unique temporal behaviors and operational constraints. To replicate these real-world dynamics, we apply channel-specific downsampling: solar-related channels are resampled at 20-minute intervals, capturing smooth and gradual irradiance trends, while wind-related channels are resampled at 5-minute intervals to reflect the rapid fluctuations characteristic of wind dynamics. This restructured dataset captures modality-specific asynchrony and sampling heterogeneity, offering a challenging benchmark for assessing model robustness under realistic conditions where different signal types operate on distinct temporal resolutions.

**Weather-practical (Weather)**  Weather-practical is constructed from the widely used Weather (Wu et al., 2021) benchmark. While the original dataset assumes uniform sampling periods across all channels, this assumption rarely holds in real-world environmental monitoring systems. In Weather-practical, we reorganize the channels to reflect heterogeneous sampling frequencies, based on the intrinsic temporal dynamics of each channel. For instance, fast-changing environmental channels such as wind velocity, precipitation, and solar radiation are sampled at 10-minute intervals, whereas more stable channels such as air pressure or vapor pressure are sampled at a coarser hourly rate. Mid-range channels such as air temperature and specific humidity are sampled at 20-minute or 30-minute intervals in accordance with their moderate temporal variability. This configuration, summarized in Table 7, mimics the asynchronous and heterogeneous acquisition patterns encountered in operational weather stations and provides a more realistic testbed for evaluating forecasting models under non-uniform input conditions.

---

[1]http://www.nemweb.com.au/

Table 7: Channel Specification in the **Weather-practical** Dataset

| Channel | Description | Sampling Period |
|---------|-------------|-----------------|
| p (mbar) | Air Pressure | 1 hour |
| T (°C) | Air Temperature | 20 min |
| Tpot (K) | Potential Temperature | 1 hour |
| Tdew (°C) | Dew Point Temperature | 1 hour |
| rh (%) | Relative Humidity | 30 min |
| VPmax (mbar) | Saturation Water Vapor Pressure | 1 hour |
| VPact (mbar) | Actual Water Vapor Pressure | 1 hour |
| VPdef (mbar) | Water Vapor Pressure Deficit | 1 hour |
| sh (g/kg) | Specific Humidity | 1 hour |
| H2OC (mmol/mol) | Water Vapor Concentration | 1 hour |
| rho (kg/m$^3$) | Air Density | 1 hour |
| wv (m/s) | Wind Velocity | 10 min |
| max. wv (m/s) | Max Wind Velocity | 10 min |
| wd (°) | Wind Direction | 10 min |
| rain (mm) | Precipitation | 10 min |
| raining (s) | Duration of Precipitation | 10 min |
| SWDR (W/m$^2$) | Short Wave Downward Radiation | 10 min |
| PAR ($\mu$mol/m$^2$/s) | Photosynthetically Active Radiation | 10 min |
| max. PAR ($\mu$mol/m$^2$/s) | Max PAR | 10 min |
| Tlog (°C) | Internal Logger Temperature | 1 hour |
| OT | Operational Timestamp (Offset) | 10 min |

**EPA-Air (EPA)**  EPA-Air is sourced from air quality monitoring data (Chang et al., 2025) of the U.S. Environmental Protection Agency[2], collects measurements from several monitoring regions across the United States. In real-world air quality monitoring, different channels are reported at heterogeneous temporal resolutions due to channel-specific measurement practices. To reflect these conditions, we assign channel-specific sampling periods: temperature-related variables are recorded at 1-hour intervals, fine particulate matter ($PM_{2.5}$) is reported at 8-hour intervals, air quality indices (AQI) are provided at daily intervals, and ozone measurements are reported at weekly intervals. For our experiments, we selected four representative cities, Maricopa, Richmond, Los Angeles, and Hillsborough, from the original dataset that contains measurements from eight metropolitan areas. This real-world dataset preserves the inherent asynchrony and heterogeneity of environmental sensing, yielding a benchmark that mirrors practical conditions where meteorological variables, pollutant measurements, and derived indices are recorded on distinct temporal resolutions.

**LNG Cargo Handling System (CHS)**  CHS is a real-world dataset collected from an LNG (Liquefied Natural Gas) carrier during a ballast voyage. The dataset contains time series measurements from a total of 52 channels, each sampled at distinct frequencies based on their physical characteristics and operational requirements. Specifically, weather and navigation channels, such as latitude and wave period, are sourced from external systems and sampled at a coarse hourly rate. In contrast, onboard sensor channels are sampled more frequently to support real-time monitoring and control. These include machinery-related channels (e.g., Low Duty Compressor (LDC) suction pressure, main engine inlet pressure, and LNG consumption), sampled at 10-minute intervals, and cargo tank sensor channels (e.g., tank temperature at top, mid, and bottom levels, and tank pressure), sampled at 1-minute intervals. For experimental evaluation, we selected 10 key channels from the full set of 52, focusing on those most relevant to navigation and weather conditions, cargo tank thermodynamics, and machinery operations.

The LNG carrier is equipped with four independent cargo tanks. During voyage, ambient heat ingress into the cargo tanks causes the LNG stored in these tanks to partially vaporize, producing what is known as Boil-Off Gas (BOG). Efficient BOG management is critical: the gas can be used as fuel, reliquefied, or combusted via a Gas Combustion Unit (GCU), which converts it into $CO_2$ before discharge. The operational objective is to minimize unnecessary combustion while maximizing energy utilization as fuel, all while maintaining tank pressure within safe operational bounds. The

---

[2]https://www.epa.gov/data

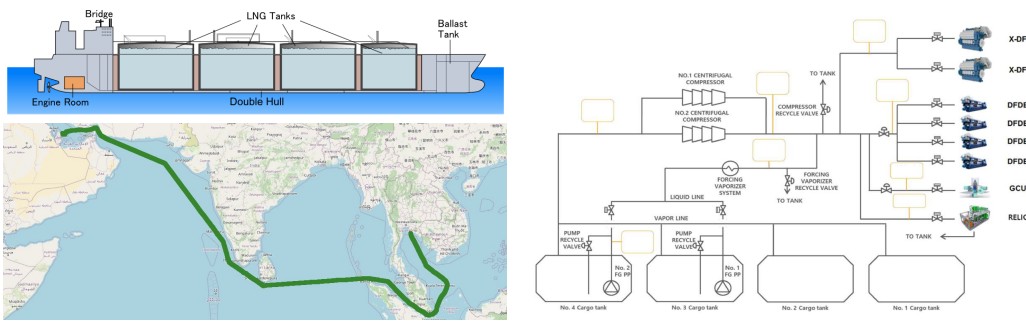

(a) LNG Carrier (from Wikimedia) and Route    (b) Diagram of LNG Cargo Handling System

Figure 5: Overview of the LNG Cargo Handling System (CHS).

CHS dataset captures the dynamic interplay among external conditions, cargo tank behavior, and onboard machinery responses. It provides a foundation for predictive modeling of key control channels, including tank gas pressure and LDC suction pressure. Accurate prediction of these channels is important for safe and energy-efficient voyage execution under the channel-wise asynchronous sampling conditions found in real-world LNG cargo operations.

## A.2 IMPLEMENTATION DETAILS

**Model Hyperparameters and GPU Setup** All experiments were conducted using a single NVIDIA RTX 3090 GPU with 24GB memory. We used the Adam optimizer across all models, with initial learning rates per dataset for fair comparison: $10^{-4}$ for ETT1, ETT2, SolarWind and EPA, $10^{-2}$ for Weather, and $10^{-3}$ for the CHS dataset. Training was run for a maximum of 10 epochs with early stopping based on validation performance. For models with a stackable encoder block, the number of encoder layers was set to 2. For Transformer-based models, the model dimension $d_{\text{model}}$ was searched over $\{128, 256, 512\}$, and the feedforward expansion ratio $d_{\text{ff}}$ / $d_{\text{model}}$ was selected from $\{1, 2, 4\}$. Details of the hyperparameters used for each dataset are fully specified in the provided code repository [3]. We adopted most baseline implementations from the TimesNet (Wu et al., 2023) repository. DUET (Qiu et al., 2025), TimeFilter (Hu et al., 2025), CrossGNN (Huang et al., 2023), t-PatchGNN (Zhang et al., 2024), Hi-Patch (Luo et al., 2025), BiTGraph (Chen et al., 2024b), originally provided in a separate repository, were reimplemented and integrated into the TimesNet framework for consistency. The hyperparameters of these baselines were aligned with those specified in their original repositories. In our proposed ChannelTokenFormer (CTF), the number of channel tokens was selected from $\{1, 2, 3\}$, with the optimal value determined separately for each dataset.

**Training and Evaluation under Our Proposed Practical Setting** For regular time series baseline models, both training and testing were conducted on linearly interpolated data due to architectural constraints. To ensure a fair and consistent comparison, we preserved the original design of the baselines and trained them on interpolated inputs with MSE loss. In contrast, our proposed model, ChannelTokenFormer (CTF), is explicitly designed to operate on non-interpolated inputs that reflect the original, variable sampling periods of each channel. To ensure uniform input shape compatible with existing architectures, we apply forward-fill interpolation during dataset preprocessing; however, this step serves merely as a structural tool to meet input dimensionality requirements, not as a modeling assumption. During the channel-wise patch embedding phase, all forward-filled values are explicitly excluded: only the valid indices $L_i$, corresponding to truly observed data points, are used for computation. For irregular and missing handling time series baselines, we followed their original frameworks by providing an input mask, which was used for both data processing and MSE computation. This procedure is exactly consistent with the forward-fill interpolation–based approach. To ensure fair evaluation, both the baselines and our model compute channel-aggregated metrics (CMSE and CMAE, as defined in Section 3) only over the valid indices $H_i$, which correspond to actually observed target values. Thus, although the input preprocessing methods differ, all models are

---

[3]Code: https://github.com/jinkwan1115/ChannelTokenFormer.git

evaluated on the same set of ground truth targets $(y_1, \ldots, y_{H_i})$. Note that under the practical setting, the number of valid test samples is inherently reduced, since the test set is subsampled according to the maximum sampling factor across channels. In summary, each model is trained under settings aligned with its architectural requirements, but all are evaluated uniformly based on their accuracy in predicting truly observed values. Some recent methods adopt a "drop-last" trick (Qiu et al., 2024) to improve performance. In our experiments, we do not apply this operation.

**Input/Output Length Settings per Dataset**   As described in Section 5, we set the input and prediction lengths for each dataset to reflect the temporal characteristics of the underlying signals. Instead of fixing the number of time steps across all channels, we define these lengths over a consistent time duration to accommodate channel-wise variations in both sampling period and dominant frequency components. For implementation and reporting convenience, however, we represent input and output lengths in terms of the number of steps corresponding to the channel with the smallest sampling period. The input and prediction lengths (in time steps, relative to the finest resolution channel) for each dataset are as follows: **ETT1**, **ETT2**: 192 (input) → [192, 336, 768, 1152] (prediction), **SolarWind**: 576 → [288, 576, 864, 1152], **Weather**: 144 → [144, 288, 576, 864], **EPA**: 96 → [96, 192, 288, 384], **CHS**: 120 → [120, 240, 480, 840]. Additional details such as patch length for each channel in each public dataset are fully specified in the released code files.

**Details of Channel-wise Frequency-based Dynamic Patching**   In *dominant frequency detection via Fast Fourier Transform (FFT)*, each input channel is represented as a one-dimensional sequence $\mathbf{x}^{(i)}$, and its amplitude spectrum $\mathbf{A}^{(i)} = |\hat{\mathbf{x}}^{(i)}|$ is computed. Instead of relying on a predefined set of candidate periods, the dominant period is searched *within a reasonable range (empirically up to $\sim 200$)*, which covers common real-world cycles such as hourly, daily, and weekly periodicities. The frequency index $k_i^*$ with the highest amplitude in this range is identified and converted into the corresponding dominant period $p_i^* = \frac{L_i}{k_i^*}$. If the amplitude $\mathbf{A}^{(i)}$ at $k_i^*$ exceeds a fixed threshold (empirically set to the 70th percentile of $\mathbf{A}^{(i)}$), $p_i^*$ is regarded as a strong dominant period. In *patch length assignment with relative sampling periods*, the relative sampling period of channel $i$ is defined as $r_i = \frac{s_i}{s_{min}}$, where $s_i$ is the actual sampling period and $s_{min}$ is the minimum across all channels. The patch length $l_i$, representing the number of points per patch, is then assigned as $l_i = \lfloor p_i^*/r_i \rfloor$, ensuring consistency with both the detected periodicity and the sampling resolution. In *the fallback mechanism*, if no dominant frequency passes the 70th percentile threshold, a sampling-aware patching rule is applied that directly determines patch lengths based on each channel's sampling rate, without enforcing periodicity-based constraints. This approach is designed for cases where periodicity is too weak or noisy to guide patching, and it prioritizes maintaining consistency with the intrinsic resolution of each channel.

**Example of Channel-wise Frequency-based Dynamic Patching**   For example, in the SolarWind dataset, wind power is recorded every 5 minutes and solar power every 20 minutes. Over a 2-day period, this corresponds to 576 time steps for wind power (resulting patch length: 48 steps) and 144 for solar power (resulting patch length: 18 steps). Assuming patching is performed based on each channel's dominant period, this yields 576 / 48 = 12 local tokens for wind power and 144 / 18 = 8 for solar power. Detailed calculations are provided below.

- Relative sampling periods: $r_1 = 1, r_2 = 4$
- FFT-detected dominant periods: $p_1^* = 48, p_2^* = 72$
- Resulting patch lengths: $l_1 = \lfloor p_1^*/r_1 \rfloor = \lfloor 48/1 \rfloor = 48$ , $l_2 = \lfloor p_2^*/r_2 \rfloor = \lfloor 72/4 \rfloor = 18$
- Number of patches:
  - $P_{ch1} = \lfloor L/l_1 \rfloor = \lfloor 576/48 \rfloor = 12$
  - $P_{ch2} = \lfloor L/(l_2 \cdot r_2) \rfloor = \lfloor 576/(18 \cdot 4) \rfloor = 8$

**Algorithmic Details of Frequency-based Dynamic Patching and Tokenization**   Algorithm 1 implements the frequency-based dynamic patching and tokenization procedure described below. Given that all channels are observed over the same time interval, we first compute the relative sampling period of each channel with respect to the smallest sampling period across channels. This determines how many samples (i.e., its effective sequence length) fall into this interval. For each channel, we then set the patch size primarily based on its dominant period. In practice, if the dominant period

---

**Algorithm 1** Frequency-based Dynamic Patching and Tokenization

---

**Require:** Multivariate series $X \in \mathbb{R}^{N \times L}$; sampling periods $\{s_i\}_{i=1}^N$; dominant periods $\{D_i\}_{i=1}^N$; horizon $H$;
    mask prob. $q$; #channel tokens $m$
**Ensure:** Token sequence $S$
1:   $s_{\min} \leftarrow \min_{i \in [N]} s_i$
2:   $S \leftarrow []$                                                        ▷ final token list
3:   **for** $i = 1$ to $N$ **do**
4:      $s_i' \leftarrow s_i / s_{\min}$                                     ▷ relative sampling period
5:      $L_i \leftarrow L / s_i'$                                        ▷ valid length of channel $i$
6:      $p_i \leftarrow D_i / s_i'$                                     ▷ patch length in samples
7:      $K_i \leftarrow L_i / p_i = L / D_i$                         ▷ number of patches
8:      Partition $X^{(i)} \in \mathbb{R}^{L_i}$ into $\{P_j^{(i)}\}_{j=1}^{K_i}$, where $P_j^{(i)} \in \mathbb{R}^{p_i}$
9:      $T^{(i)} \leftarrow []$                                 ▷ local tokens kept after masking
10:     **for** $j = 1$ to $K_i$ **do**
11:        **if** $P_j^{(i)}$ is fully unobserved (test time) **or** $\text{Bernoulli}(q) = 1$ (training) **then**     ▷ patch masking
12:          **continue**
13:        **end if**
14:        $T_j^{(i)} \leftarrow W_{p_i} P_j^{(i)} + e_{i,j}^{\text{pos}}$                 ▷ $W_{p_i} \in \mathbb{R}^{d \times p_i}$, $T_j^{(i)} \in \mathbb{R}^d$
15:        Append $T_j^{(i)}$ to $T^{(i)}$
16:     **end for**
17:     Initialize learnable channel tokens $\{C_k^{(i)}\}_{k=1}^m$, $C_k^{(i)} \in \mathbb{R}^d$
18:     $S^{(i)} \leftarrow [T^{(i)}, C_1^{(i)}, \ldots, C_m^{(i)}] + e_i^{\text{chn}}$         ▷ $e_i^{\text{chn}}$ is broadcast to all tokens
19:     Append $S^{(i)}$ to $S$
20: **end for**
21: **return** $S$

---

cannot be reliably estimated for a channel, one can instead define its patch size using only the relative sampling period, i.e., a sampling-aware patching rule. The resulting series for each channel is then partitioned into patches, and we discard patches that are either entirely unobserved during test time or randomly skipped with probability $q$ during training, so that the token sequence reflects only informative segments. Each remaining patch is linearly projected and enriched with positional embeddings, after which a small set of learnable channel tokens is appended and a shared channel embedding is added to all tokens from the same channel. This yields a unified token sequence that encodes heterogeneous sampling periods and channel-wise frequency characteristics without relying on interpolation.

**Test-time Missing Conditions** For controlled missingness injection, we vary the overall missing ratio and apply block-wise masking by independently sampling contiguous blocks for each channel and each sample. This procedure mimics naturally occurring block-wise gaps caused by sensor malfunctions or maintenance events in real deployments. It produces a range of challenging, randomly located missing blocks that are applied identically to all compared methods at test time.

To specifically evaluate robustness under structured contiguous missing conditions, we additionally introduce block-wise missing intervals into the test-time inputs of the SolarWind dataset. Each channel is assigned randomly positioned missing blocks, with each block spanning exactly one patch length. This ensures that at least one patch per channel is entirely unobserved, enabling a direct assessment of our model's patch-level masking strategy at test time. During training, our model applies random patch masking, as shown in Table 8, training with a 0.4 random masking ratio consistently improved model robustness across different missing ratios.

## B   SPECTRAL DISTORTION INDUCED BY INTERPOLATION

### B.1   BACKGROUND

In practical time series forecasting scenarios, multivariate signals often feature **channel-wise asynchronous sampling periods** across channels, stemming from both inter-system and intra-system heterogeneity in sensor design and operation. However, most existing forecasting models assume a *regular and complete* time grid, necessitating the use of interpolation to align all channels to a

Table 8: Average forecasting performance of our CTF model on the SolarWind dataset, evaluated under block-wise missing at test time with varying missing ratios $m$, and trained with different patch masking ratios $t$.

| Missing ratio (Train) | $t = 0.1$ | | $t = 0.2$ | | $t = 0.3$ | | $t = 0.4$ | | $t = 0.5$ | |
|---|---|---|---|---|---|---|---|---|---|---|
| Metric | CMSE | CMAE | CMSE | CMAE | CMSE | CMAE | CMSE | CMAE | CMSE | CMAE |
| $m = 0.125$ | 0.420 | 0.472 | 0.414 | 0.466 | 0.413 | 0.466 | **0.408** | **0.463** | 0.412 | 0.466 |
| $m = 0.250$ | 0.458 | 0.504 | 0.444 | 0.491 | 0.439 | 0.488 | 0.428 | 0.482 | **0.427** | **0.481** |
| $m = 0.375$ | 0.490 | 0.537 | 0.474 | 0.521 | 0.464 | 0.515 | **0.452** | **0.508** | 0.452 | 0.509 |
| $m = 0.500$ | 0.527 | 0.573 | 0.498 | 0.549 | 0.488 | 0.541 | **0.474** | **0.533** | 0.474 | 0.536 |

common temporal resolution prior to model ingestion. While this preprocessing step ensures input compatibility, it inevitably introduces **spectral distortion** (Feng et al., 2004; Fitzgerald & Anderson, 1992) by artificially synthesizing intermediate values, thereby altering the original frequency characteristics of the signals. Interpolation imposes artificial continuity by estimating unobserved values from nearby observations, which can fundamentally reshape both the temporal dynamics and spectral content of the signal. This appendix provides a theoretical explanation of how such interpolation-induced distortion arises in the frequency domain, as observed through the Fast Fourier Transform (FFT).

### B.2 LINEAR INTERPOLATION-INDUCED FREQUENCY DOMAIN DISTORTION

Let $x(t)$ denote a real-valued time series obtained via uniform but sparse sampling relative to other channels. Let $\tilde{x}(t)$ denote the version of $x(t)$ with intermediate values filled in via linear interpolation between uniformly sampled points. This interpolation process can be interpreted as a convolution in the time domain, as linear interpolation is equivalent to applying a piecewise linear (triangular) filter to the sampled signal. By the Convolution Theorem, this results in a multiplication in the frequency domain:

$$\tilde{X}(f) = X(f) \cdot H(f),$$

where $H(f)$ is the frequency response of the triangular interpolation kernel, given by $\text{sinc}^2(f)$. This imposes a low-pass filtering effect that attenuates high-frequency components of the original signal, and constitutes the primary source of spectral distortion introduced by linear interpolation. Such distortion manifests in multiple forms: (i) amplitude attenuation within the Nyquist limit, due to low-pass filtering that suppresses mid-to-high frequency energy; (ii) amplitude gaps, due to the non-uniform and oscillatory nature of the interpolation kernel's frequency response, which causes broad spectral suppression in the mid-to-high frequency range; (iii) spectral leakage beyond the Nyquist frequency, caused by the non-ideal frequency response of finite-support interpolation kernels; and (iv) phase delays, introduced by the filtering process, especially in the mid-to-high frequency range, and observable in the FFT phase spectrum.

**(1) Amplitude Attenuation**  Under linear interpolation, the reconstructed signal exhibits reduced overall spectral energy, particularly in the mid-to-high frequency range:

$$\|\tilde{X}(f)\|_2^2 < \|X(f)\|_2^2.$$

This occurs because each frequency component is scaled by a factor $|H(f)| < 1$ for $f > 0$, resulting in suppressed spectral magnitude:

$$|\tilde{X}(f)| = |X(f)| \cdot |H(f)| < |X(f)|.$$

Attenuation becomes stronger at higher frequencies, leading to a net reduction in total spectral energy and a smoother appearance in the time domain. This loss of high-frequency information can impair the model's ability to capture fine-grained temporal patterns in forecasting tasks.

**(2) Amplitude Gaps**  The frequency response of linear interpolation exhibits oscillatory behavior due to its squared sinc form, resulting in repeated dips and low-gain regions across the spectrum, particularly in the mid-to-high frequency range. When these regions coincide with frequencies that

contain substantial energy in the original signal, the interpolated spectrum displays significant local suppression—observable as amplitude gaps:

$$|\tilde{X}(f)| \ll |X(f)| \quad \text{for } f \in [f_{\text{mid}}, f_{\text{high}}].$$

Such non-uniform attenuation alters the spectral envelope and can adversely impact the performance of forecasting models.

**(3) Spectral Leakage (High-Frequency Artifacts)**   The frequency response of linear interpolation, given by $H(f) = \text{sinc}^2(f)$, is not band-limited and extends well beyond the Nyquist frequency, with *decaying side lobes* across the spectrum. As a result, interpolation artificially introduces high-frequency components into the signal, even when the original data is strictly band-limited. This is particularly evident in the FFT magnitude spectrum(see Figure 6a), where the interpolated signal exhibits spectral energy in regions beyond the Nyquist limit. These high-frequency artifacts are not part of the true signal but emerge from the spectral spreading effect caused by the interpolation kernel. They distort the original spectral structure by injecting spurious out-of-band energy, which can mislead models.

**(4) Phase Delay**   Although linear interpolation produces a smooth and continuous waveform from discretely sampled data, it introduces a frequency-dependent shift in the phase spectrum. Let $\angle X(f)$ and $\angle \tilde{X}(f)$ denote the phase spectra of the original and interpolated signals, respectively. Then the phase difference is given by:

$$\Delta\phi(f) = \angle \tilde{X}(f) - \angle X(f) \leq 0,$$

which remains negative across most frequencies, with the delay becoming notably apparent from the mid-frequency band onward (see Figure 6b). This indicates that the interpolated signal lags behind the original in phase. The delay is negligible at low frequencies but grows rapidly in the high-frequency band, due to the squared sinc frequency response of linear interpolation. When the signal's dominant spectral components lie in the high-frequency region, this phase delay leads to substantial misalignment of key waveform features such as peaks and transitions. Even if the interpolated waveform appears visually smooth in the time domain, the underlying temporal displacement of high-frequency content can significantly degrade forecasting accuracy, especially in tasks that rely on precise timing of local patterns.

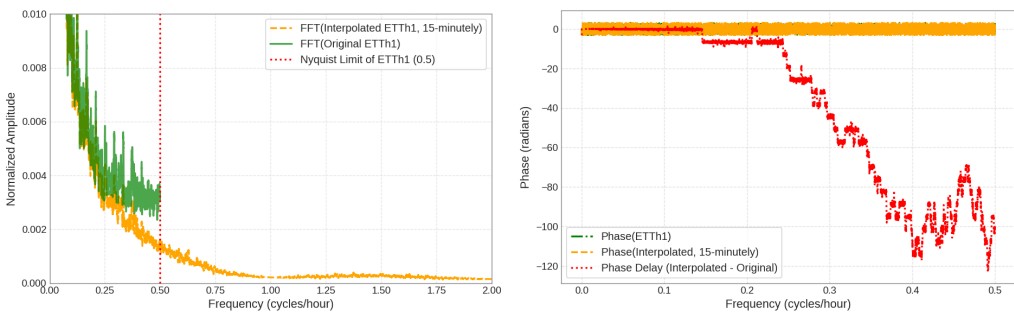

(a) FFT Magnitude Spectrum and Distortion          (b) FFT Phase Spectrum and Delay

Figure 6: Spectral distortion caused by linear interpolation in the ETTh1 dataset. The left panel shows amplitude attenuation and spectral leakage in the magnitude spectrum, while the right panel illustrates phase delay in the FFT phase spectrum.

## B.3   Summary and Implications for Forecasting Performance

The four types of distortion (amplitude attenuation, amplitude gaps, spectral leakage, and phase delay) collectively degrade the spectral integrity of interpolated signals. These artifacts can mislead forecasting models into learning spurious temporal or frequency patterns. The issue becomes more severe in multivariate settings with channel-wise asynchronous sampling, where aligning all channels to a uniform time grid via interpolation introduces artificial synchrony and continuity. To mitigate such distortion, we advocate for **interpolation-free modeling**, where models are designed

to operate directly on asynchronously sampled and non-uniformly aligned signals, without relying on synthesized values that risk obscuring the true structure of the original signals.

Input-level distortions act as a primary source of frequency bias and, in turn, forecasting error. Thus, the key priority is not to design ever more elaborate frequency-domain heads on top of already interpolated inputs, but rather to adopt an interpolation-free architecture that can ingest asynchronously sampled, partially missing multivariate series without forcing them onto an artificially regular grid. In practice, we observe that frequency-based approaches such as FreTS and FITS are vulnerable to such frequency biases, showing significant performance degradation on real-world datasets with strong channel-wise asynchrony (see Table 18 for detailed results). Moreover, the frequency-bias analysis comparing CTF and TimeXer, which are fundamentally similar except for CTF's interpolation-free module, indicates that removing interpolation yields more faithful spectral characteristics relative to the original time series and, consequently, improved forecasts (see Section 6.3). CTF is designed with exactly this objective in mind.

## C  FURTHER ANALYSIS

### C.1  ADDITIONAL ABLATIONS

To complement the main experiments, we provide additional ablation studies highlighting the contribution of key architectural components in ChannelTokenFormer (CTF). As shown in Table 9, incorporating *learnable channel-specific embeddings* consistently improves performance across all datasets. This suggests that explicitly encoding channel identity through channel embeddings provides a strong inductive bias, particularly in heterogeneous multivariate settings. Furthermore, as shown in Table 10, *setting patch lengths to match each channel's periodic characteristics*, rather than using a fixed patch length, has a clear impact on forecasting performance. This underscores the importance of reflecting channel-specific sampling periodicities when handling asynchronous time series.

Table 9: Effect of channel embedding on forecasting performance. Results are reported in CMSE, averaged over all prediction lengths. Overall, channel embedding improves performance on most datasets, indicating its effectiveness in modeling channel-specific dynamics.

| Setting / Dataset | ETT1 | ETT2 | SolarWind | Weather | EPA | CHS |
|---|---|---|---|---|---|---|
| CTF with channel embedding | **0.398** | **0.377** | 0.400 | **0.269** | **0.775** | **0.282** |
| CTF w/o channel embedding | 0.404 | 0.379 | **0.399** | 0.283 | 0.781 | 0.285 |

Table 10: Impact of patch length on forecasting performance. Results are reported in CMSE, averaged over all prediction lengths. In general, frequency-guided patching improves performance on most datasets, suggesting that adapting patch lengths to each channel's periodic structure is beneficial.

| Patch length / Dataset | ETT1 | ETT2 | SolarWind | Weather | EPA | CHS |
|---|---|---|---|---|---|---|
| FFT-based | **0.398** | **0.377** | 0.400 | **0.274** | **0.775** | **0.282** |
| 8 | 0.404 | 0.378 | 0.405 | 0.289 | 0.817 | 0.300 |
| 16 | 0.401 | 0.378 | 0.400 | 0.289 | 0.807 | 0.301 |
| 32 | 0.424 | 0.383 | **0.399** | 0.306 | 0.799 | 0.307 |

To further examine the effectiveness of our design beyond what was demonstrated in Section 5.2, we conduct an extended ablation study focusing on the three practical challenges of multivariate forecasting: *Dependency*, *Asynchrony*, and *Missingness*. As summarized in Table 11, we selectively disable each architectural component corresponding to the three core challenges addressed by our design, and evaluate all ablation settings under varying levels of test-time missingness. To remove *Channel Dependence*, we eliminate attention between channel tokens from different channels. To

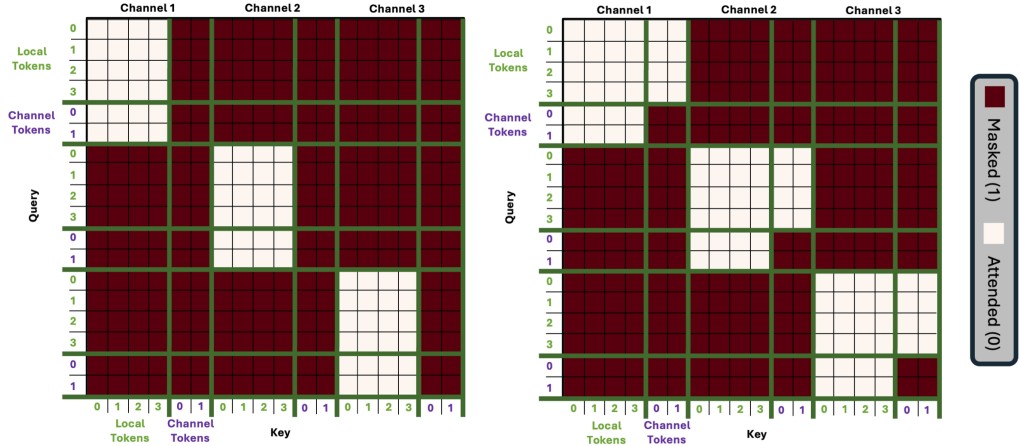

Figure 7: Visualization of the attention masks used in **Channel-Independent (CI)** masking strategies. **CI-ReadOnly** (left) allows only unidirectional attention from channel tokens to local tokens within the same channel, while **CI-Mutual** (right) permits bidirectional attention between local and channel tokens. In both cases, no cross-channel attention is allowed.

assess the impact of *Dynamic Patching*, we prevent the model from incorporating dominant frequency information extracted from FFT as well as channel-specific sampling periods. For *Patch Masking*, we disable the masking mechanism for block-wise missing intervals during both training and testing. Across all settings, the full model consistently achieves the best results, confirming that each architectural element contributes complementarily to the model's overall robustness. Among these, patch masking emerges as critical for handling contiguous missing blocks, while channel dependency and dynamic patching offer additional gains under asynchronous and heterogeneous conditions.

Table 11: Results of the ablation study under channel-wise asynchronous sampling with block-wise test-time missing. Results are shown for different missing ratios $m$, with metrics averaged across all prediction lengths on the SolarWind dataset.

| Test-time Missing Ratio | $m = 0.125$ | | $m = 0.250$ | | $m = 0.375$ | | $m = 0.500$ | |
|---|---|---|---|---|---|---|---|---|
| Ablation Setting | CMSE | CMAE | CMSE | CMAE | CMSE | CMAE | CMSE | CMAE |
| **Full model (CTF)** | **0.408** | **0.463** | **0.428** | **0.482** | **0.452** | **0.508** | **0.474** | **0.533** |
| w/o Channel Dependence | 0.408 | 0.463 | 0.429 | 0.484 | 0.474 | 0.521 | 0.522 | 0.551 |
| w/o Dynamic patching | 0.452 | 0.496 | 0.474 | 0.515 | 0.494 | 0.536 | 0.517 | 0.562 |
| w/o Patch masking | 0.412 | 0.465 | 0.434 | 0.484 | 0.458 | 0.508 | 0.482 | 0.533 |

## C.2 COMPARISON OF ATTENTION MASKING STRATEGIES

We conduct an ablation study comparing six variants of attention masking strategies within our unified attention layer. To avoid analytical redundancy, all strategies share a common constraint: *channel tokens are prohibited from attending to any other tokens within the same channel, including themselves*. This restriction eliminates redundant intra-channel interactions and encourages richer cross-channel communication.

The six masking strategies fall into two categories: **Channel-Independent (CI)** and **Channel-Dependent (CD)**. The key distinction lies in whether cross-channel attention among channel tokens is permitted. CD strategies are further subdivided based on the scope and indexing of allowed inter-channel connections. Table 12 presents comparative results for the masking strategies on the SolarWind dataset (Case 2).

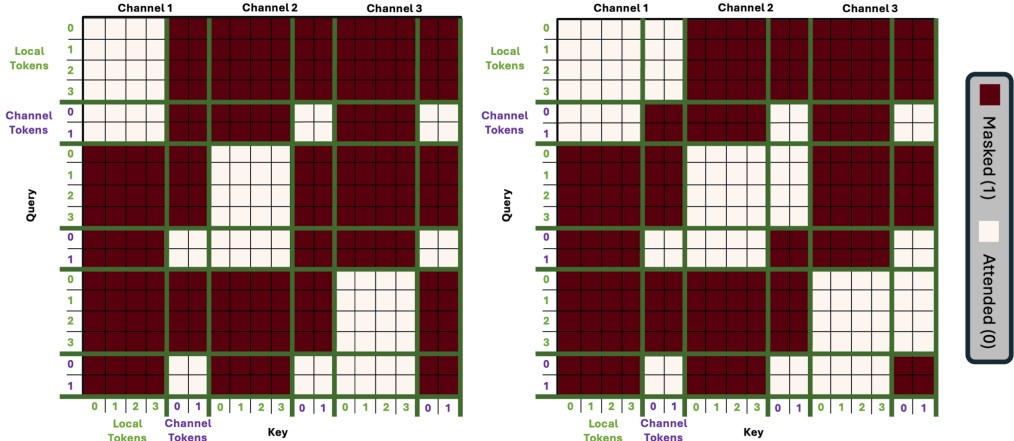

Figure 8: Visualization of the attention masks used in **Channel-Dependent (CD)** masking strategies with global cross-channel token attention. **CD-ReadOnly** (left) extends CI-ReadOnly by allowing channel tokens to attend to channel tokens from other channels, while **CD-Mutual** (right) permits bidirectional attention within channels and additionally enables global cross-channel attention among channel tokens.

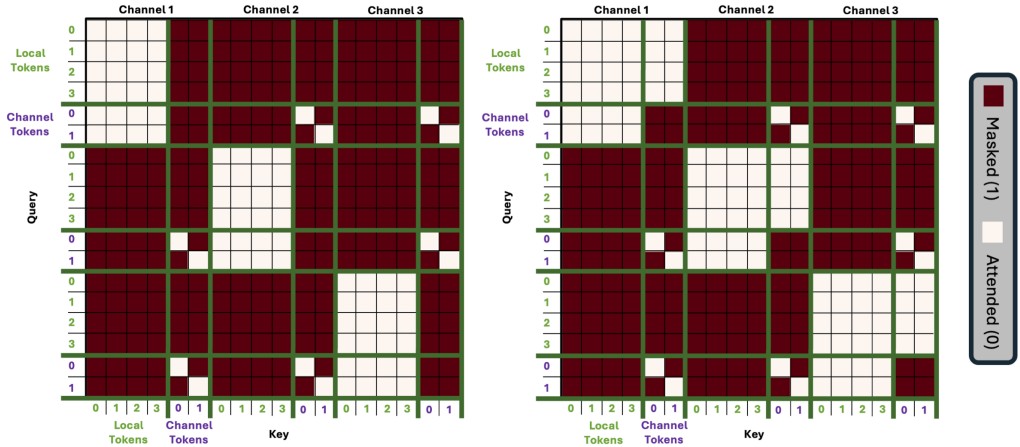

Figure 9: Visualization of the attention masks used in **Indexed Channel-Dependent (CD)** masking strategies. **CD-ReadOnly-Indexed** (left) and **CD-Mutual-Indexed** (right) restrict cross-channel attention to channel tokens with the same index across different channels. This design promotes role-specific specialization while maintaining intra-channel locality for local tokens.

- **Channel-Independent Masking (2 variants):** All attention operations are confined to within each channel. No cross-channel interaction is allowed (see Figure 7).
    - **CI-ReadOnly**: Channel tokens can attend to local tokens (read-only summarization), but not vice versa.
    - **CI-Mutual**: Local and channel tokens within a channel can attend to each other bidirectionally.
- **Channel-Dependent Masking (4 variants)**: Local tokens remain strictly within-channel, attending only to tokens from the same channel. In contrast, channel tokens are permitted to interact across channels, subject to specific masking rules.

- **Global Cross-Channel Token Attention:** Each channel token can attend to all channel tokens from other channels, but not to those from its own channel. This facilitates rich cross-channel communication while maintaining strict inter-channel exclusivity (see Figure 8).
    * **CD-ReadOnly**: Extends CI-ReadOnly by allowing channel tokens to also attend to channel tokens from other channels. This allows each channel token to summarize both intra- and inter-channel information without influencing local token representations directly.
    * **CD-Mutual**: Extends CI-Mutual with additional global attention among channel tokens across different channels. It enables two-way communication between channel tokens, potentially enhancing global coordination and information exchange across channels.

- **Indexed Cross-Channel Token Attention**: When multiple channel tokens are assigned per channel, attention is restricted to those with matching indices across channels (e.g., index-0 tokens attend only to other index-0 tokens). This design encourages index-wise specialization and alignment (see Figure 9).
    * **CD-ReadOnly-Indexed**: Read-only variant allowing same-index channel tokens to attend to each other across channels. This selective communication enables controlled abstraction across semantically aligned channel positions.
    * **CD-Mutual-Indexed**: Mutual variant enabling bidirectional intra-channel and same-index inter-channel attention. It balances local coherence and cross-channel consistency by unifying both local and indexed global interactions.

Table 12: Comparison of attention masking strategies on the SolarWind dataset under the block-wise test-time missing scenario. Results are reported in CMSE and CMAE, averaged over all prediction lengths with a missing ratio $m = 0.375$. Among the methods, **CD-ReadOnly** and **CD-ReadOnly-Indexed** achieve the best performance under our proposed practical setting.

| Strategy | CI-ReadOnly | CI-Mutual | CD-ReadOnly | CD-Mutual | CD-ReadOnly-Indexed | CD-Mutual-Indexed |
|---|---|---|---|---|---|---|
| CMSE | 0.474 | 0.457 | **0.452** | 0.456 | **0.452** | 0.457 |
| CMAE | 0.521 | 0.512 | **0.508** | 0.510 | **0.508** | 0.512 |

## C.3 SCALABILITY ANALYSIS

We further investigate the scalability of ChannelTokenFormer (CTF) with respect to both the **number of channels** and the **input sequence length**.

**Channel scalability.** We measured runtime per training iteration and maximum memory usage as the number of input channels increases. Results in Table 13 show that CTF scales reliably up to 275 channels on a single 24GB GPU. Out-of-memory (OOM) is observed only at 280 channels. Training runtime grows proportionally with channel count (e.g., 2.69s/iter at $C = 200$), *while inference latency remains comparable to TimeXer (0.903s vs. 0.917s).*

Table 13: Channel scalability of CTF on a single GPU with 24GB VRAM. We report training runtime (s/iter) and peak memory usage (MB) as the number of input channels increases. CTF scales reliably up to 275 channels, with out-of-memory (OOM) occurring at 280 channels.

| Metric / # of Channels | 5 | 30 | 50 | 100 | 200 | 275 | 280 |
|---|---|---|---|---|---|---|---|
| Runtime (s/iter) | 0.0186 | 0.1193 | 0.2307 | 0.7173 | 2.6912 | 5.4175 | OOM |
| Memory (MB) | 108.9 | 460.3 | 911.2 | 2843.9 | 10386.1 | 17834.9 | – |

**Input length scalability.** We also tested longer input horizons. As shown in Table 14, inference runtime remains stable (0.014s → 0.016s) as sequence length grows from 96 to 2048. Memory usage scales linearly with input length.

Table 14: Input length scalability of CTF. We report training runtime (s/iter), peak memory usage (MB), and inference latency as input length grows. Inference runtime remains stable (0.014s → 0.016s), while memory usage scales linearly with length.

| Metric / Input Length | 96 | 512 | 1024 | 2048 |
|---|---|---|---|---|
| Training Runtime (s/iter) | 0.028 | 0.048 | 0.074 | 0.147 |
| Training Memory (MB) | 126.1 | 403.3 | 945.8 | 2921.4 |
| Inference Runtime (s/iter) | 0.0140 | 0.0147 | 0.0147 | 0.0159 |
| Inference Memory (MB) | 126.1 | 404.8 | 945.8 | 2921.4 |

**Summary.** These results demonstrate that the masking-based attention design in CTF can handle high-density multivariate forecasting and long-horizon inputs with stable inference latency. While memory grows linearly and limits appear at very high channel counts, the model remains efficient in practical regimes.

## C.4 COMPARISON WITH MODIFIED TIMEXER FOR ASYNCHRONOUS SAMPLING

For TimeXer (Wang et al., 2024e), we incorporated channel-wise asynchronously sampled inputs, along with their corresponding sampling information, as exogenous inputs, following the same strategy (shared tokenizers) used in our model. To ensure consistency and reduce redundancy, channels with identical sampling periods shared the same modeling configuration when each was used as the forecasting target. We conducted experiments on the ETT1, ETT2, and SolarWind datasets for Case 1 (Section 5.1), and on the SolarWind dataset for Case 2 (Section 5.1). Results are reported as CMSE and CMAE averaged over four prediction horizons. As summarized in Table 15, our CTF achieves consistently better or comparable performance compared to the modified TimeXer variants.

Table 15: Comparison of CTF with modified TimeXer under asynchronous sampling. Results are reported in CMSE and CMAE averaged over four prediction horizons.

| Model | | **CTF (ours)** | | TimeXer (mod.) | | TimeXer | |
|---|---|---|---|---|---|---|---|
| Dataset / Metric | | CMSE | CMAE | CMSE | CMAE | CMSE | CMAE |
| Case 1 | ETT1 | **0.398** | **0.411** | 0.404 | 0.416 | 0.420 | 0.424 |
| | ETT2 | 0.377 | 0.383 | **0.374** | **0.382** | 0.381 | 0.389 |
| | SolarWind | **0.400** | **0.448** | 0.417 | 0.450 | 0.420 | 0.468 |
| Case 2 | SolarWind | **0.440** | **0.496** | 0.474 | 0.504 | 0.462 | 0.504 |

## C.5 COMPARISON WITH FREQUENCY-BASED APPROACHES

As representative frequency-based approaches, we compare FreTS (Yi et al., 2023b) and FITS (Xu et al., 2024) against our CTF on a range of datasets exhibiting realistic channel-wise asynchrony. Because these methods operate primarily in the frequency domain, they are expected to be vulnerable to frequency biases introduced by interpolation-based preprocessing; indeed, the experimental results (Table 18) show significant performance degradation under strong inter-channel asynchrony, particularly on EPA and CHS, whereas CTF remains substantially more robust.

## C.6 ANALYSIS OF HANDLING DISCRETE MISSINGNESS

First, the real-world benchmarks we use already contain discrete missingness. For example, in ETTm (15-minutely) data, one can observe that missing entries are implicitly encoded as zeros. Not only our model but also other baselines rely on a small neural tokenizer (patch embedding network) to process these inputs. We empirically observe that this stage already provides a reasonable degree of robustness to discrete missing values. Figure 10 illustrates this effect. Even when 30% of entries are converted into discrete missing values, the resulting patch-embedding cloud in t-SNE space

remains closely aligned with that of the clean inputs. This behavior is consistent with the fact that the patch tokenizer is a shallow neural network (e.g., a small MLP-based projector) that implements a relatively smooth, approximately Lipschitz-continuous mapping from input patches to embeddings. Under such mappings, localized perturbations that affect only a small fraction of time steps such as isolated missing entries induce only bounded changes in the embedding space, rather than causing drastic geometric distortions (Fazlyab et al., 2019; Zhang et al., 2022a). Consequently, discrete missingness tends to be absorbed as small, localized noise at the tokenizer level, which helps explain why downstream forecasting performance is largely preserved even when a nontrivial fraction of discrete missing values is present.

To evaluate robustness more precisely under discrete missing conditions at test time, we introduce discrete missing values into the test-time inputs of two datasets, ETT1 and EPA. In both datasets, we randomly insert zero-filled gaps of short, varying lengths (5 to 20 time steps), which are substantially smaller than the patch size, into each channel. These settings are designed to test model robustness to scattered missing values. For ETT1 with discrete missingness, we compare CTF against standard time-series baselines, including strong CD and CI approaches. For EPA with discrete missingness, we select two representative baselines, Hi-Patch (Luo et al., 2025) and ContiFormer (Chen et al., 2023). Since ContiFormer is an ODE-based model with prohibitively long training times, we reimplement it within the TimesNet framework by approximating its ODE solver, following the structure of TimeXer (Wang et al., 2024e), to ensure both efficiency and consistency. However, because this reimplemented ContiFormer does not natively conform to the irregular-sampling frameworks considered and operates on interpolated inputs, we treat it under the same procedure as the regular baselines. We focus on two challenging monitoring sites, Los Angeles (LA) and Hillsborough, to evaluate robustness under more difficult real-world conditions. Experimental results under this setting are reported in Table 16 and Table 17. Overall, we observe that most models exhibit only minor degradation under discrete missingness for small missing ratios (0.1–0.3). Across all tested missing ratios, however, CTF consistently maintains stronger performance than the compared methods on both datasets.

Table 16: Average CMSE and CMAE for all prediction lengths of ETT1 dataset under short-range test-time missing intervals with varying missing ratios $m$. CTF is trained with a random patch masking ratio of 0.4 to enhance robustness against missing inputs.

| Approach | Channel-Dependent | | | | | | | Channel-Independent | | |
|---|---|---|---|---|---|---|---|---|---|---|
| Model | CTF(ours) | TimeFilter | DUET | TimeXer | iTrans. | CrossGNN | TimesNet | TimeMixer++ | PatchTST | DLinear |
| Metric | CMSE CMAE | CMSE CMAE | CMSE CMAE | CMSE CMAE | CMSE CMAE | CMSE CMAE | CMSE CMAE | CMSE CMAE | CMSE CMAE | CMSE CMAE |
| $m = 0.1$ | 0.415 0.424 | **0.412 0.418** | 0.445 0.442 | 0.440 0.441 | 0.455 0.449 | 0.448 0.434 | 0.490 0.468 | 0.449 0.448 | 0.439 0.438 | 0.453 0.442 |
| $m = 0.2$ | 0.434 0.438 | **0.433 0.438** | 0.475 0.465 | 0.466 0.461 | 0.477 0.467 | 0.471 0.452 | 0.508 0.486 | 0.474 0.469 | 0.465 0.458 | 0.483 0.465 |
| $m = 0.3$ | **0.453 0.451** | 0.455 0.453 | 0.506 0.489 | 0.495 0.481 | 0.500 0.483 | 0.495 0.470 | 0.527 0.502 | 0.499 0.486 | 0.491 0.478 | 0.514 0.486 |
| $m = 0.4$ | **0.471 0.463** | 0.474 0.466 | 0.572 0.530 | 0.526 0.502 | 0.523 0.499 | 0.520 0.487 | 0.550 0.519 | 0.522 0.503 | 0.517 0.497 | 0.544 0.506 |
| $m = 0.5$ | **0.493 0.477** | 0.498 0.484 | 0.595 0.545 | 0.563 0.526 | 0.548 0.515 | 0.549 0.506 | 0.576 0.536 | 0.546 0.518 | 0.547 0.518 | 0.577 0.526 |

## D  FORECASTING RESULTS IN A CONVENTIONAL MULTIVARIATE SETTING

To demonstrate that our model remains effective even under standard multivariate cases, we additionally evaluate it on regularly sampled multivariate time series without any missing values. By aligning the sampling period and patch length across all channels, our model can be directly applied to this conventional setting. This evaluation shows that our model remains applicable beyond the proposed practical settings. For comparison, baseline results (Hu et al., 2025; Qiu et al., 2025; Chen et al., 2025; Wang et al., 2024e;a; Liu et al., 2024c; Huang et al., 2023; Li et al., 2023b; Nie et al., 2023; Zhang & Yan, 2023; Das et al., 2023; Wu et al., 2023; Zeng et al., 2023) are obtained either from the original publications or by reproducing them using the TSLib codebase (Wang et al., 2024d). We compare forecasting performance on five widely used benchmark datasets (Zhou et al., 2021; Wu et al., 2021): ETTh1, ETTh2, ETTm1, ETTm2, and Weather. As shown in Table 19, our model achieves performance comparable to or better than state-of-the-art baselines across all benchmark datasets in the conventional forecasting setting.

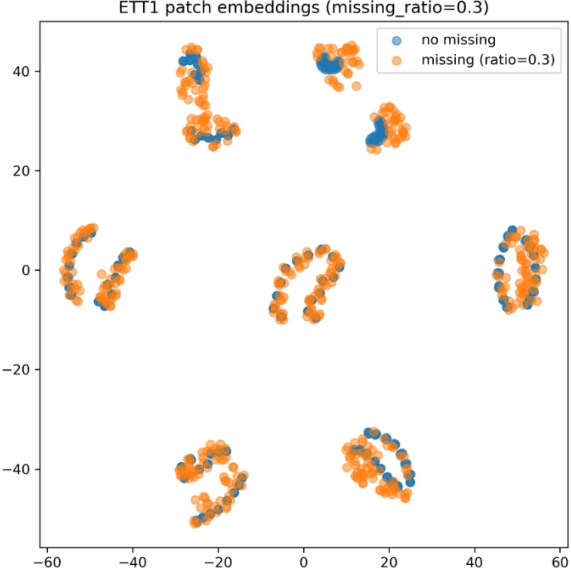

Figure 10: t-SNE visualization of patch embeddings produced by CTF's patch tokenizer for a batch of ETT1 test samples under two conditions: with 30% discrete missing values injected and with no injected missingness. The near-overlap between the two embedding distributions suggests that this patch embedding stage already provides a reasonable degree of robustness to discrete missing values.

Table 17: Average CMSE and CMAE for all prediction lengths of EPA dataset (LA, Hillsborough) under short-range test-time missing intervals with varying missing ratios $m$. CTF is trained with a random patch masking ratio of 0.4 to enhance robustness against missing inputs.

| Model | CTF (ours) | | Hi-Patch | | ContiFormer | |
|---|---|---|---|---|---|---|
| Metric | CMSE | CMAE | CMSE | CMAE | CMSE | CMAE |
| $m = 0.1$ | **1.074** | **0.702** | 1.084 | 0.742 | 1.219 | 0.744 |
| $m = 0.2$ | 1.146 | **0.731** | **1.121** | 0.754 | 1.237 | 0.757 |
| $m = 0.3$ | 1.226 | **0.756** | **1.204** | 0.781 | 1.254 | 0.773 |
| $m = 0.4$ | **1.290** | **0.788** | 1.383 | 0.839 | 1.298 | 0.801 |

Table 18: Forecasting performance on the channel-wise asynchrony compared with frequency-based approaches.

| Model | CTF (ours) | | FreTS | | FITS | |
|---|---|---|---|---|---|---|
| Metric | CMSE | CMAE | CMSE | CMAE | CMSE | CMAE |
| ETT1 | **0.399** | **0.410** | 0.436 | 0.432 | 0.443 | 0.430 |
| ETT2 | **0.377** | **0.383** | 0.440 | 0.428 | 0.391 | 0.389 |
| SolarWind | **0.403** | **0.452** | 0.416 | 0.502 | 0.485 | 0.481 |
| Weather | **0.275** | **0.296** | 0.285 | 0.328 | 0.297 | 0.304 |
| EPA | **0.776** | **0.586** | 1.018 | 0.684 | 1.245 | 0.762 |
| CHS | **0.285** | 0.126 | 0.485 | 0.344 | 0.289 | **0.121** |

Table 19: Full results of the long-term forecasting task in the conventional setting. The best results are highlighted in **bold**, and the second-best results are underlined.

| Model | | CTF(ours) | | TimeMixer++ | | TimeFilter | | DUET | | SimpleTM | | TimeXer | | iTrans. | | CrossGNN | | RLinear | | PatchTST | | Crossformer | | TiDE | | TimesNet | | DLinear | |
|---|---|---|---|---|---|---|---|---|---|---|---|---|---|---|---|---|---|---|---|---|---|---|---|---|---|---|---|---|---|
| | Metric | CMSE | CMAE | CMSE | CMAE | CMSE | CMAE | CMSE | CMAE | CMSE | CMAE | CMSE | CMAE | CMSE | CMAE | CMSE | CMAE | CMSE | CMAE | CMSE | CMAE | CMSE | CMAE | CMSE | CMAE | CMSE | CMAE | CMSE | CMAE |
| ETTh1 | 96 | 0.371 | 0.397 | 0.361 | 0.403 | 0.370 | 0.394 | 0.377 | 0.393 | 0.366 | 0.392 | 0.382 | 0.403 | 0.386 | 0.405 | 0.382 | 0.398 | 0.386 | 0.395 | 0.414 | 0.419 | 0.423 | 0.448 | 0.479 | 0.464 | 0.384 | 0.402 | 0.386 | 0.400 |
| | 192 | 0.417 | 0.426 | 0.416 | 0.441 | 0.413 | 0.420 | 0.429 | 0.425 | 0.422 | 0.421 | 0.429 | 0.435 | 0.441 | 0.436 | 0.427 | 0.425 | 0.437 | 0.424 | 0.460 | 0.445 | 0.471 | 0.474 | 0.525 | 0.492 | 0.436 | 0.429 | 0.437 | 0.432 |
| | 336 | 0.451 | 0.448 | 0.430 | 0.434 | 0.450 | 0.440 | 0.471 | 0.446 | 0.440 | 0.438 | 0.468 | 0.448 | 0.487 | 0.458 | 0.465 | 0.445 | 0.479 | 0.446 | 0.501 | 0.466 | 0.570 | 0.546 | 0.565 | 0.515 | 0.491 | 0.469 | 0.481 | 0.459 |
| | 720 | 0.462 | 0.470 | 0.467 | 0.451 | 0.448 | 0.457 | 0.496 | 0.480 | 0.463 | 0.462 | 0.469 | 0.461 | 0.503 | 0.491 | 0.472 | 0.468 | 0.481 | 0.470 | 0.500 | 0.488 | 0.653 | 0.621 | 0.594 | 0.558 | 0.521 | 0.500 | 0.519 | 0.516 |
| | Avg | 0.425 | 0.435 | 0.419 | 0.432 | 0.422 | 0.428 | 0.420 | 0.428 | 0.443 | 0.436 | 0.437 | 0.437 | 0.454 | 0.447 | 0.437 | 0.434 | 0.446 | 0.434 | 0.469 | 0.454 | 0.529 | 0.522 | 0.541 | 0.507 | 0.458 | 0.450 | 0.456 | 0.452 |
| ETTh2 | 96 | 0.288 | 0.340 | 0.276 | 0.328 | 0.283 | 0.337 | 0.296 | 0.345 | 0.281 | 0.338 | 0.286 | 0.338 | 0.297 | 0.349 | 0.309 | 0.359 | 0.288 | 0.338 | 0.302 | 0.348 | 0.745 | 0.584 | 0.400 | 0.440 | 0.340 | 0.374 | 0.333 | 0.387 |
| | 192 | 0.376 | 0.396 | 0.342 | 0.379 | 0.362 | 0.392 | 0.368 | 0.389 | 0.355 | 0.387 | 0.363 | 0.389 | 0.380 | 0.400 | 0.390 | 0.406 | 0.374 | 0.390 | 0.388 | 0.400 | 0.877 | 0.656 | 0.528 | 0.509 | 0.402 | 0.414 | 0.477 | 0.476 |
| | 336 | 0.427 | 0.433 | 0.346 | 0.398 | 0.404 | 0.424 | 0.411 | 0.422 | 0.365 | 0.401 | 0.414 | 0.423 | 0.428 | 0.432 | 0.426 | 0.444 | 0.415 | 0.426 | 0.426 | 0.433 | 1.043 | 0.731 | 0.643 | 0.571 | 0.452 | 0.452 | 0.594 | 0.541 |
| | 720 | 0.428 | 0.444 | 0.392 | 0.415 | 0.407 | 0.433 | 0.412 | 0.434 | 0.413 | 0.436 | 0.408 | 0.432 | 0.427 | 0.445 | 0.444 | 0.420 | 0.440 | 0.431 | 0.446 | 0.431 | 1.104 | 0.763 | 174 | 0.679 | 0.462 | 0.468 | 131 | 0.657 |
| | Avg | 0.380 | 0.403 | 0.339 | 0.380 | 0.364 | 0.397 | 0.372 | 0.397 | 0.353 | 0.391 | 0.367 | 0.396 | 0.383 | 0.407 | 0.393 | 0.413 | 0.374 | 0.398 | 0.387 | 0.407 | 0.942 | 0.684 | 0.611 | 0.550 | 0.414 | 0.427 | 0.559 | 0.515 |
| ETTm1 | 96 | 0.317 | 0.356 | 0.310 | 0.334 | 0.313 | 0.354 | 0.324 | 0.354 | 0.321 | 0.361 | 0.318 | 0.356 | 0.334 | 0.368 | 0.335 | 0.373 | 0.355 | 0.376 | 0.329 | 0.367 | 0.404 | 0.426 | 0.364 | 0.387 | 0.338 | 0.375 | 0.345 | 0.372 |
| | 192 | 0.359 | 0.381 | 0.348 | 0.362 | 0.356 | 0.380 | 0.369 | 0.379 | 0.360 | 0.380 | 0.362 | 0.383 | 0.387 | 0.391 | 0.372 | 0.390 | 0.391 | 0.392 | 0.367 | 0.385 | 0.450 | 0.451 | 0.398 | 0.404 | 0.374 | 0.387 | 0.380 | 0.389 |
| | 336 | 0.389 | 0.404 | 0.376 | 0.391 | 0.386 | 0.402 | 0.404 | 0.402 | 0.390 | 0.404 | 0.395 | 0.407 | 0.426 | 0.420 | 0.403 | 0.411 | 0.424 | 0.415 | 0.399 | 0.410 | 0.532 | 0.515 | 0.428 | 0.425 | 0.410 | 0.411 | 0.413 | 0.413 |
| | 720 | 0.447 | 0.440 | 0.440 | 0.423 | 0.452 | 0.437 | 0.463 | 0.437 | 0.454 | 0.438 | 0.452 | 0.441 | 0.491 | 0.459 | 0.461 | 0.442 | 0.487 | 0.450 | 0.454 | 0.439 | 0.666 | 0.589 | 0.487 | 0.461 | 0.478 | 0.450 | 0.474 | 0.453 |
| | Avg | 0.378 | 0.395 | 0.369 | 0.378 | 0.377 | 0.393 | 0.390 | 0.393 | 0.381 | 0.396 | 0.382 | 0.397 | 0.407 | 0.410 | 0.393 | 0.404 | 0.414 | 0.407 | 0.387 | 0.400 | 0.513 | 0.496 | 0.419 | 0.419 | 0.400 | 0.406 | 0.403 | 0.407 |
| ETTm2 | 96 | 0.173 | 0.255 | 0.170 | 0.245 | 0.169 | 0.255 | 0.174 | 0.255 | 0.173 | 0.257 | 0.171 | 0.256 | 0.180 | 0.264 | 0.176 | 0.266 | 0.182 | 0.265 | 0.175 | 0.259 | 0.287 | 0.366 | 0.207 | 0.305 | 0.187 | 0.267 | 0.193 | 0.292 |
| | 192 | 0.238 | 0.299 | 0.229 | 0.291 | 0.235 | 0.299 | 0.243 | 0.302 | 0.238 | 0.299 | 0.237 | 0.299 | 0.250 | 0.309 | 0.240 | 0.307 | 0.246 | 0.304 | 0.241 | 0.302 | 0.414 | 0.492 | 0.290 | 0.364 | 0.249 | 0.309 | 0.284 | 0.362 |
| | 336 | 0.295 | 0.336 | 0.303 | 0.343 | 0.293 | 0.336 | 0.304 | 0.341 | 0.296 | 0.338 | 0.296 | 0.338 | 0.311 | 0.348 | 0.300 | 0.345 | 0.307 | 0.342 | 0.305 | 0.343 | 0.597 | 0.542 | 0.377 | 0.422 | 0.321 | 0.351 | 0.369 | 0.427 |
| | 720 | 0.392 | 0.394 | 0.373 | 0.399 | 0.390 | 0.393 | 0.399 | 0.397 | 0.393 | 0.395 | 0.392 | 0.394 | 0.412 | 0.407 | 0.406 | 0.400 | 0.407 | 0.398 | 0.402 | 0.400 | 1.730 | 1.042 | 0.558 | 0.524 | 0.408 | 0.403 | 0.554 | 0.522 |
| | Avg | 0.274 | 0.321 | 0.269 | 0.320 | 0.272 | 0.321 | 0.280 | 0.324 | 0.275 | 0.322 | 0.274 | 0.322 | 0.288 | 0.332 | 0.282 | 0.330 | 0.286 | 0.327 | 0.281 | 0.326 | 0.757 | 0.610 | 0.358 | 0.404 | 0.291 | 0.333 | 0.350 | 0.401 |
| Weather | 96 | 0.159 | 0.207 | 0.155 | 0.205 | 0.153 | 0.199 | 0.163 | 0.202 | 0.162 | 0.207 | 0.157 | 0.205 | 0.174 | 0.214 | 0.159 | 0.218 | 0.192 | 0.232 | 0.177 | 0.218 | 0.158 | 0.230 | 0.202 | 0.261 | 0.172 | 0.220 | 0.196 | 0.255 |
| | 192 | 0.207 | 0.251 | 0.201 | 0.245 | 0.202 | 0.246 | 0.218 | 0.252 | 0.208 | 0.248 | 0.204 | 0.247 | 0.221 | 0.254 | 0.211 | 0.266 | 0.240 | 0.271 | 0.225 | 0.259 | 0.206 | 0.277 | 0.242 | 0.298 | 0.219 | 0.261 | 0.237 | 0.296 |
| | 336 | 0.263 | 0.292 | 0.237 | 0.265 | 0.260 | 0.289 | 0.274 | 0.294 | 0.263 | 0.290 | 0.261 | 0.290 | 0.278 | 0.296 | 0.267 | 0.310 | 0.292 | 0.307 | 0.278 | 0.297 | 0.272 | 0.335 | 0.287 | 0.335 | 0.280 | 0.306 | 0.283 | 0.335 |
| | 720 | 0.343 | 0.344 | 0.312 | 0.334 | 0.342 | 0.341 | 0.349 | 0.343 | 0.340 | 0.341 | 0.340 | 0.341 | 0.358 | 0.347 | 0.352 | 0.362 | 0.364 | 0.353 | 0.354 | 0.348 | 0.398 | 0.418 | 0.351 | 0.386 | 0.365 | 0.359 | 0.345 | 0.381 |
| | Avg | 0.243 | 0.273 | 0.226 | 0.262 | 0.239 | 0.269 | 0.251 | 0.273 | 0.243 | 0.271 | 0.241 | 0.271 | 0.258 | 0.278 | 0.247 | 0.289 | 0.272 | 0.291 | 0.259 | 0.281 | 0.259 | 0.315 | 0.271 | 0.320 | 0.259 | 0.287 | 0.265 | 0.317 |

## E  FULL RESULTS OF OUR PROPOSED PRACTICAL SETTING

Table 20 shows the results for Case 1 using asynchronous practical datasets, where datasets with channel-wise asynchronous sampling were linearly interpolated onto a regular sampling grid for

the regular time series baselines. Table 21 presents the results for Case 2, which adds block-wise test-time missing inputs on top of asynchronous sampling. All results include error bars based on 5 random seeds to reflect performance variability. Across both cases, our model consistently demonstrates stronger performance than baseline methods.

## F    Forecast Visualization

Figure 11a illustrates the forecasts of four models including our CTF on channel 3 of the ETT1 dataset under Case 1 in Section 5. The prediction horizon shows periodic sharp drops and recoveries, which are critical for accurate forecasting. Our model closely tracks these rapid transitions, while baseline models tend to underpredict their magnitude. We attribute this to the interpolation-free nature of our model, which avoids the frequency biases introduced by linear interpolation.

We also examine Case 2 in Section 5, which involves both asynchronous sampling and block-wise test time missing. Figure 11b presents a visualization of model predictions for the solar power channel from the SolarWind dataset. In this sample, a missing block of length 144 is present in the middle of the test input. This causes TimeMixer++ and TimeXer to produce noticeably noisy forecasts. Although CrossGNN shows relatively higher robustness, its predictions exhibit an underestimation of the periodic transitions. In contrast, our model accurately identifies the missing region and applies masking, effectively preventing the injection of unreliable inputs. As a result, it successfully captures both the timing and magnitude of the periodic transitions, even across the missing block.

## G    Discussion

To jointly capture temporal dynamics and cross-channel dependencies under practical settings, we proposed a unified attention masking strategy that merges intra- and inter-channel attention into a single attention layer. This design avoids the need for separate attention modules, reducing the number of parameters and simplifying the overall model architecture. This parameter efficiency is particularly beneficial for datasets with limited data, as it reduces the risk of overfitting while maintaining the model's ability to capture essential temporal and cross-channel patterns. However, this unified structure introduces scalability challenges when applied to high-dimensional datasets. As the number of channels increases, the attention layer must process a quadratic number of token interactions within a single computation step, leading to substantial computational overhead and slower training. A promising direction to mitigate this issue is to optimize the attention computation, for example, by explicit channel sparsification or grouped channel representations.

Beyond scalability, another limitation lies in the objective function. Although our model demonstrates strong predictive performance across diverse practical settings, it still suffers from amplitude attenuation, especially over longer forecasting horizons, due to the use of a standard mean squared error loss. To address this, it is crucial to explore alternative loss functions that better preserve spectral characteristics and periodicity, rather than relying solely on point-wise accuracy. This issue also extends to evaluation. Conventional metrics such as MSE may fail to penalize overly smoothed forecasts, often favoring models that suppress meaningful high-frequency or transient patterns. As a result, structurally inaccurate predictions may still appear favorable under MSE, particularly in long-term forecasting scenarios dominated by coarse trends. Refining evaluation metrics to better capture structural fidelity is therefore essential. Designing both loss functions and evaluation protocols that align with the characteristics of asynchronous, partially observed time series remains an important and open research challenge.

Table 20: Full error bars for the channel-wise asynchronous long-term multivariate forecasting task.

| Approach | | | | | | | Channel-Dependent | | | | | | | | | | | | Channel-Independent | | | | | | Irregular modeling | | | |
|---|---|---|---|---|---|---|---|---|---|---|---|---|---|---|---|---|---|---|---|---|---|---|---|---|---|---|---|---|---|
| Model | | CTF(ours) | | TimeFilter | | DUET | | TimeXer | | iTrans. | | CrossGNN | | TimesNet | | TimeMixer++ | | PatchTST | | DLinear | | Hi-Patch | | tPatchGNN | |
| Metric | | CMSE | CMAE | CMSE | CMAE | CMSE | CMAE | CMSE | CMAE | CMSE | CMAE | CMSE | CMAE | CMSE | CMAE | CMSE | CMAE | CMSE | CMAE | CMSE | CMAE | CMSE | CMAE | CMSE | CMAE |
| ETT1 | 192 | 0.335 ±0.001 | 0.370 ±0.001 | 0.340 ±0.003 | 0.371 ±0.003 | 0.355 ±0.001 | 0.382 ±0.001 | 0.354 ±0.001 | 0.381 ±0.001 | 0.356 ±0.002 | 0.384 ±0.002 | 0.355 ±0.001 | 0.373 ±0.001 | 0.388 ±0.007 | 0.407 ±0.007 | 0.363 ±0.001 | 0.389 ±0.005 | 0.343 ±0.001 | 0.374 ±0.001 | 0.352 ±0.001 | 0.373 ±0.001 | 0.396 ±0.025 | 0.422 ±0.008 | 0.413 ±0.015 | 0.440 ±0.009 |
| | 336 | 0.364 ±0.001 | 0.390 ±0.000 | 0.376 ±0.001 | 0.396 ±0.001 | 0.390 ±0.001 | 0.405 ±0.001 | 0.394 ±0.001 | 0.409 ±0.001 | 0.396 ±0.001 | 0.410 ±0.001 | 0.393 ±0.002 | 0.396 ±0.001 | 0.414 ±0.011 | 0.424 ±0.005 | 0.399 ±0.003 | 0.412 ±0.001 | 0.378 ±0.002 | 0.397 ±0.001 | 0.390 ±0.001 | 0.397 ±0.001 | 0.421 ±0.013 | 0.440 ±0.005 | 0.429 ±0.009 | 0.450 ±0.008 |
| | 768 | 0.428 ±0.001 | 0.430 ±0.001 | 0.445 ±0.002 | 0.439 ±0.003 | 0.455 ±0.002 | 0.444 ±0.002 | 0.451 ±0.002 | 0.444 ±0.001 | 0.472 ±0.003 | 0.454 ±0.001 | 0.460 ±0.002 | 0.436 ±0.001 | 0.487 ±0.015 | 0.463 ±0.008 | 0.465 ±0.010 | 0.451 ±0.004 | 0.443 ±0.004 | 0.435 ±0.002 | 0.457 ±0.002 | 0.442 ±0.003 | 0.462 ±0.009 | 0.489 ±0.004 | 0.489 ±0.006 | 0.484 ±0.004 |
| | 1152 | 0.467 ±0.001 | 0.452 ±0.002 | 0.488 ±0.003 | 0.465 ±0.001 | 0.495 ±0.002 | 0.467 ±0.001 | 0.489 ±0.002 | 0.463 ±0.001 | 0.514 ±0.001 | 0.477 ±0.002 | 0.503 ±0.003 | 0.460 ±0.001 | 0.519 ±0.017 | 0.481 ±0.010 | 0.506 ±0.012 | 0.475 ±0.002 | 0.481 ±0.002 | 0.457 ±0.002 | 0.500 ±0.002 | 0.467 ±0.002 | 0.515 ±0.021 | 0.491 ±0.012 | 0.529 ±0.006 | 0.505 ±0.004 |
| ETT2 | 192 | 0.264 ±0.002 | 0.312 ±0.001 | 0.263 ±0.004 | 0.315 ±0.002 | 0.273 ±0.004 | 0.321 ±0.003 | 0.269 ±0.004 | 0.319 ±0.002 | 0.282 ±0.002 | 0.329 ±0.001 | 0.269 ±0.001 | 0.318 ±0.000 | 0.282 ±0.004 | 0.328 ±0.004 | 0.290 ±0.007 | 0.333 ±0.004 | 0.276 ±0.002 | 0.323 ±0.001 | 0.287 ±0.004 | 0.347 ±0.004 | 0.297 ±0.005 | 0.345 ±0.003 | 0.286 ±0.004 | 0.344 ±0.004 |
| | 336 | 0.319 ±0.002 | 0.349 ±0.001 | 0.325 ±0.004 | 0.353 ±0.001 | 0.324 ±0.004 | 0.353 ±0.003 | 0.327 ±0.004 | 0.355 ±0.003 | 0.349 ±0.001 | 0.368 ±0.001 | 0.329 ±0.001 | 0.357 ±0.002 | 0.341 ±0.006 | 0.363 ±0.003 | 0.351 ±0.009 | 0.373 ±0.006 | 0.339 ±0.005 | 0.365 ±0.003 | 0.367 ±0.010 | 0.400 ±0.008 | 0.344 ±0.005 | 0.370 ±0.005 | 0.342 ±0.007 | 0.378 ±0.007 |
| | 768 | 0.436 ±0.004 | 0.416 ±0.002 | 0.446 ±0.004 | 0.422 ±0.002 | 0.450 ±0.004 | 0.425 ±0.008 | 0.439 ±0.003 | 0.420 ±0.003 | 0.451 ±0.005 | 0.426 ±0.002 | 0.449 ±0.001 | 0.421 ±0.000 | 0.466 ±0.009 | 0.433 ±0.005 | 0.446 ±0.010 | 0.427 ±0.006 | 0.450 ±0.003 | 0.431 ±0.001 | 0.527 ±0.020 | 0.490 ±0.012 | 0.453 ±0.011 | 0.430 ±0.006 | 0.450 ±0.011 | 0.434 ±0.006 |
| | 1152 | 0.488 ±0.004 | 0.454 ±0.004 | 0.498 ±0.004 | 0.461 ±0.001 | 0.504 ±0.019 | 0.456 ±0.010 | 0.478 ±0.014 | 0.454 ±0.004 | 0.501 ±0.010 | 0.459 ±0.004 | 0.498 ±0.002 | 0.456 ±0.002 | 0.509 ±0.018 | 0.461 ±0.007 | 0.496 ±0.011 | 0.465 ±0.005 | 0.494 ±0.003 | 0.465 ±0.003 | 0.641 ±0.022 | 0.550 ±0.012 | 0.494 ±0.004 | 0.460 ±0.002 | 0.495 ±0.013 | 0.468 ±0.006 |
| SolarWind | 288 | 0.333 ±0.008 | 0.399 ±0.012 | 0.333 ±0.007 | 0.399 ±0.003 | 0.361 ±0.003 | 0.399 ±0.003 | 0.341 ±0.012 | 0.414 ±0.011 | 0.362 ±0.006 | 0.419 ±0.004 | 0.341 ±0.011 | 0.403 ±0.014 | 0.362 ±0.010 | 0.404 ±0.006 | 0.356 ±0.012 | 0.411 ±0.008 | 0.330 ±0.008 | 0.404 ±0.005 | 0.343 ±0.005 | 0.450 ±0.005 | 0.372 ±0.007 | 0.438 ±0.005 | 0.378 ±0.006 | 0.451 ±0.009 |
| | 576 | 0.400 ±0.005 | 0.452 ±0.006 | 0.398 ±0.014 | 0.445 ±0.011 | 0.435 ±0.011 | 0.484 ±0.009 | 0.424 ±0.007 | 0.473 ±0.008 | 0.453 ±0.004 | 0.478 ±0.010 | 0.438 ±0.010 | 0.463 ±0.010 | 0.474 ±0.036 | 0.476 ±0.021 | 0.427 ±0.010 | 0.464 ±0.008 | 0.416 ±0.010 | 0.469 ±0.006 | 0.418 ±0.003 | 0.513 ±0.018 | 0.433 ±0.022 | 0.475 ±0.018 | 0.449 ±0.017 | 0.499 ±0.019 |
| | 864 | 0.428 ±0.009 | 0.472 ±0.008 | 0.434 ±0.020 | 0.471 ±0.015 | 0.468 ±0.016 | 0.517 ±0.013 | 0.452 ±0.021 | 0.487 ±0.017 | 0.511 ±0.014 | 0.512 ±0.007 | 0.508 ±0.011 | 0.505 ±0.007 | 0.500 ±0.016 | 0.497 ±0.012 | 0.457 ±0.016 | 0.493 ±0.010 | 0.452 ±0.018 | 0.495 ±0.015 | 0.453 ±0.005 | 0.542 ±0.004 | 0.447 ±0.010 | 0.485 ±0.008 | 0.469 ±0.019 | 0.508 ±0.018 |
| | 1152 | 0.453 ±0.009 | 0.484 ±0.013 | 0.450 ±0.013 | 0.480 ±0.008 | 0.488 ±0.015 | 0.534 ±0.014 | 0.478 ±0.012 | 0.503 ±0.008 | 0.552 ±0.013 | 0.531 ±0.006 | 0.573 ±0.010 | 0.543 ±0.009 | 0.547 ±0.012 | 0.519 ±0.012 | 0.475 ±0.014 | 0.504 ±0.006 | 0.471 ±0.008 | 0.498 ±0.005 | 0.470 ±0.001 | 0.557 ±0.001 | 0.471 ±0.008 | 0.494 ±0.005 | 0.494 ±0.011 | 0.514 ±0.009 |
| Weather | 144 | 0.185 ±0.001 | 0.231 ±0.000 | 0.181 ±0.005 | 0.224 ±0.001 | 0.231 ±0.002 | 0.269 ±0.004 | 0.223 ±0.004 | 0.255 ±0.004 | 0.235 ±0.002 | 0.272 ±0.002 | 0.209 ±0.014 | 0.247 ±0.011 | 0.209 ±0.014 | 0.248 ±0.009 | 0.178 ±0.002 | 0.225 ±0.002 | 0.175 ±0.014 | 0.207 ±0.000 | 0.207 ±0.000 | 0.262 ±0.000 | 0.218 ±0.018 | 0.263 ±0.017 | 0.233 ±0.016 | 0.272 ±0.011 |
| | 288 | 0.247 ±0.002 | 0.279 ±0.002 | 0.249 ±0.009 | 0.276 ±0.005 | 0.284 ±0.003 | 0.305 ±0.004 | 0.275 ±0.004 | 0.294 ±0.003 | 0.289 ±0.003 | 0.308 ±0.004 | 0.257 ±0.005 | 0.297 ±0.007 | 0.288 ±0.024 | 0.305 ±0.014 | 0.248 ±0.013 | 0.279 ±0.007 | 0.247 ±0.011 | 0.315 ±0.015 | 0.261 ±0.001 | 0.309 ±0.001 | 0.277 ±0.019 | 0.306 ±0.017 | 0.288 ±0.011 | 0.308 ±0.007 |
| | 576 | 0.316 ±0.001 | 0.324 ±0.000 | 0.315 ±0.005 | 0.322 ±0.005 | 0.345 ±0.002 | 0.344 ±0.001 | 0.333 ±0.003 | 0.332 ±0.002 | 0.349 ±0.001 | 0.344 ±0.002 | 0.324 ±0.006 | 0.339 ±0.006 | 0.361 ±0.013 | 0.354 ±0.010 | 0.323 ±0.003 | 0.326 ±0.002 | 0.318 ±0.006 | 0.359 ±0.009 | 0.322 ±0.001 | 0.358 ±0.002 | 0.337 ±0.012 | 0.342 ±0.011 | 0.346 ±0.011 | 0.345 ±0.006 |
| | 864 | 0.352 ±0.000 | 0.351 ±0.001 | 0.349 ±0.004 | 0.350 ±0.004 | 0.372 ±0.002 | 0.364 ±0.002 | 0.368 ±0.006 | 0.359 ±0.004 | 0.380 ±0.024 | 0.369 ±0.002 | 0.367 ±0.011 | 0.378 ±0.011 | 0.396 ±0.024 | 0.381 ±0.016 | 0.354 ±0.004 | 0.355 ±0.004 | 0.359 ±0.000 | 0.393 ±0.000 | 0.356 ±0.005 | 0.394 ±0.007 | 0.372 ±0.011 | 0.369 ±0.009 | 0.383 ±0.000 | 0.372 ±0.000 |
| EPA | 96 | 0.641 ±0.007 | 0.525 ±0.003 | 0.620 ±0.005 | 0.492 ±0.004 | 0.613 ±0.006 | 0.499 ±0.003 | 0.651 ±0.025 | 0.509 ±0.007 | 0.657 ±0.030 | 0.519 ±0.013 | 0.725 ±0.031 | 0.558 ±0.013 | 0.720 ±0.013 | 0.553 ±0.008 | 0.705 ±0.030 | 0.551 ±0.019 | 0.624 ±0.004 | 0.495 ±0.002 | 0.921 ±0.013 | 0.657 ±0.004 | 0.673 ±0.005 | 0.570 ±0.004 | 0.663 ±0.009 | 0.563 ±0.004 |
| | 192 | 0.782 ±0.005 | 0.591 ±0.002 | 0.858 ±0.006 | 0.602 ±0.002 | 0.795 ±0.002 | 0.586 ±0.003 | 0.873 ±0.010 | 0.612 ±0.004 | 0.888 ±0.005 | 0.618 ±0.002 | 0.949 ±0.024 | 0.661 ±0.007 | 0.922 ±0.024 | 0.639 ±0.009 | 0.923 ±0.020 | 0.640 ±0.010 | 0.856 ±0.007 | 0.602 ±0.003 | 1.054 ±0.004 | 0.716 ±0.001 | 0.811 ±0.006 | 0.634 ±0.003 | 0.796 ±0.009 | 0.629 ±0.004 |
| | 288 | 0.825 ±0.006 | 0.603 ±0.001 | 0.951 ±0.004 | 0.636 ±0.002 | 0.843 ±0.005 | 0.606 ±0.002 | 0.971 ±0.008 | 0.647 ±0.003 | 0.964 ±0.005 | 0.643 ±0.001 | 1.070 ±0.052 | 0.705 ±0.014 | 1.005 ±0.007 | 0.668 ±0.004 | 1.008 ±0.017 | 0.664 ±0.005 | 0.940 ±0.007 | 0.632 ±0.002 | 1.091 ±0.009 | 0.732 ±0.002 | 0.856 ±0.006 | 0.646 ±0.003 | 0.846 ±0.016 | 0.648 ±0.005 |
| | 384 | 0.858 ±0.008 | 0.622 ±0.003 | 1.023 ±0.011 | 0.667 ±0.004 | 0.876 ±0.006 | 0.626 ±0.002 | 1.047 ±0.017 | 0.676 ±0.005 | 1.019 ±0.004 | 0.666 ±0.001 | 1.171 ±0.076 | 0.735 ±0.016 | 1.103 ±0.038 | 0.712 ±0.014 | 1.087 ±0.035 | 0.694 ±0.010 | 0.995 ±0.006 | 0.659 ±0.003 | 1.121 ±0.009 | 0.748 ±0.002 | 0.894 ±0.003 | 0.667 ±0.002 | 0.900 ±0.009 | 0.672 ±0.002 |
| CHS | 120 | 0.103 ±0.002 | 0.065 ±0.000 | 0.116 ±0.005 | 0.065 ±0.003 | 0.131 ±0.012 | 0.072 ±0.003 | 0.122 ±0.003 | 0.067 ±0.000 | 0.132 ±0.002 | 0.072 ±0.000 | 0.108 ±0.003 | 0.065 ±0.003 | 0.142 ±0.017 | 0.077 ±0.003 | 0.116 ±0.003 | 0.067 ±0.003 | 0.129 ±0.007 | 0.065 ±0.000 | 0.113 ±0.003 | 0.098 ±0.012 | 0.123 ±0.003 | 0.067 ±0.005 | 0.106 ±0.000 | 0.061 ±0.000 |
| | 240 | 0.166 ±0.001 | 0.086 ±0.000 | 0.184 ±0.010 | 0.090 ±0.002 | 0.193 ±0.007 | 0.097 ±0.004 | 0.178 ±0.003 | 0.089 ±0.000 | 0.184 ±0.006 | 0.095 ±0.004 | 0.168 ±0.000 | 0.092 ±0.007 | 0.210 ±0.015 | 0.103 ±0.003 | 0.173 ±0.003 | 0.089 ±0.002 | 0.194 ±0.010 | 0.090 ±0.002 | 0.182 ±0.004 | 0.144 ±0.009 | 0.184 ±0.002 | 0.091 ±0.001 | 0.174 ±0.006 | 0.087 ±0.003 |
| | 480 | 0.314 ±0.003 | 0.139 ±0.000 | 0.339 ±0.021 | 0.142 ±0.003 | 0.333 ±0.007 | 0.144 ±0.002 | 0.323 ±0.004 | 0.139 ±0.000 | 0.332 ±0.009 | 0.145 ±0.004 | 0.314 ±0.006 | 0.148 ±0.007 | 0.356 ±0.017 | 0.153 ±0.004 | 0.328 ±0.010 | 0.146 ±0.005 | 0.346 ±0.008 | 0.147 ±0.003 | 0.378 ±0.023 | 0.285 ±0.033 | 0.328 ±0.033 | 0.141 ±0.002 | 0.325 ±0.006 | 0.139 ±0.003 |
| | 840 | 0.555 ±0.004 | 0.216 ±0.003 | 0.576 ±0.016 | 0.217 ±0.002 | 0.572 ±0.009 | 0.217 ±0.002 | 0.570 ±0.013 | 0.217 ±0.002 | 0.573 ±0.010 | 0.215 ±0.003 | 0.553 ±0.004 | 0.232 ±0.009 | 0.613 ±0.042 | 0.225 ±0.005 | 0.566 ±0.013 | 0.215 ±0.003 | 0.591 ±0.006 | 0.226 ±0.003 | 0.731 ±0.072 | 0.466 ±0.049 | 0.571 ±0.008 | 0.216 ±0.040 | 0.574 ±0.012 | 0.213 ±0.003 |

Table 21: Full error bars for the SolarWind dataset under block-wise test-time missingness with varying missing ratios $m$.

| Approach | | Channel-Dependent | | | | | | | | | | | | Channel-Independent | | | | | | Irregular modeling | | | | Missing | |
|---|---|---|---|---|---|---|---|---|---|---|---|---|---|---|---|---|---|---|---|---|---|---|---|---|---|
| Model | | CTF(ours) | | TimeFilter | | DUET | | TimeXer | | iTrans. | | CrossGNN | | TimesNet | | TimeMixer++ | | PatchTST | | DLinear | | Hi-Patch | | tPatchGNN | | BitGraph | |
| Metric | | CMSE | CMAE | CMSE | CMAE | CMSE | CMAE | CMSE | CMAE | CMSE | CMAE | CMSE | CMAE | CMSE | CMAE | CMSE | CMAE | CMSE | CMAE | CMSE | CMAE | CMSE | CMAE | CMSE | CMAE | CMSE | CMAE |
| $m=0.125$ | 288 | 0.336±0.003 | 0.414±0.000 | 0.350±0.001 | 0.410±0.000 | 0.365±0.005 | 0.435±0.000 | 0.350±0.000 | 0.419±0.011 | 0.367±0.006 | 0.426±0.011 | 0.347±0.011 | 0.412±0.011 | 0.366±0.011 | 0.409±0.008 | 0.360±0.008 | 0.415±0.008 | 0.337±0.007 | 0.413±0.007 | 0.349±0.005 | 0.458±0.005 | 0.369±0.007 | 0.441±0.008 | 0.380±0.014 | 0.454±0.011 | 0.333±0.002 | 0.429±0.004 |
| | 576 | 0.400±0.002 | 0.456±0.002 | 0.441±0.005 | 0.473±0.007 | 0.442±0.007 | 0.493±0.007 | 0.427±0.006 | 0.477±0.008 | 0.459±0.004 | 0.484±0.004 | 0.445±0.004 | 0.472±0.000 | 0.478±0.027 | 0.479±0.018 | 0.433±0.009 | 0.470±0.018 | 0.423±0.003 | 0.479±0.007 | 0.423±0.003 | 0.522±0.005 | 0.429±0.023 | 0.475±0.023 | 0.448±0.021 | 0.501±0.021 | 0.413±0.028 | 0.497±0.028 |
| | 864 | 0.433±0.003 | 0.483±0.005 | 0.454±0.004 | 0.483±0.001 | 0.476±0.009 | 0.526±0.009 | 0.457±0.020 | 0.492±0.006 | 0.517±0.006 | 0.518±0.007 | 0.515±0.006 | 0.514±0.007 | 0.507±0.004 | 0.502±0.010 | 0.466±0.016 | 0.498±0.011 | 0.462±0.019 | 0.508±0.009 | 0.457±0.005 | 0.549±0.004 | 0.443±0.013 | 0.484±0.021 | 0.471±0.021 | 0.512±0.023 | 0.447±0.011 | 0.526±0.034 |
| | 1152 | 0.465±0.002 | 0.500±0.000 | 0.481±0.002 | 0.498±0.000 | 0.497±0.018 | 0.544±0.015 | 0.482±0.013 | 0.508±0.011 | 0.559±0.007 | 0.538±0.004 | 0.581±0.013 | 0.553±0.004 | 0.523±0.062 | 0.507±0.039 | 0.484±0.016 | 0.510±0.009 | 0.481±0.007 | 0.507±0.004 | 0.475±0.010 | 0.565±0.004 | 0.466±0.009 | 0.494±0.004 | 0.502±0.008 | 0.520±0.006 | 0.510±0.062 | 0.585±0.058 |
| $m=0.25$ | 288 | 0.350±0.001 | 0.431±0.002 | 0.362±0.005 | 0.422±0.022 | 0.387±0.015 | 0.457±0.008 | 0.358±0.011 | 0.434±0.012 | 0.389±0.007 | 0.447±0.006 | 0.359±0.004 | 0.428±0.012 | 0.379±0.012 | 0.423±0.004 | 0.389±0.016 | 0.438±0.007 | 0.361±0.010 | 0.439±0.007 | 0.361±0.004 | 0.473±0.005 | 0.381±0.007 | 0.460±0.010 | 0.390±0.015 | 0.467±0.012 | 0.343±0.008 | 0.438±0.007 |
| | 576 | 0.417±0.001 | 0.473±0.001 | 0.446±0.012 | 0.482±0.007 | 0.466±0.016 | 0.518±0.011 | 0.440±0.006 | 0.490±0.008 | 0.479±0.005 | 0.505±0.004 | 0.459±0.009 | 0.489±0.008 | 0.511±0.022 | 0.505±0.016 | 0.463±0.011 | 0.492±0.009 | 0.454±0.016 | 0.514±0.010 | 0.434±0.003 | 0.536±0.004 | 0.459±0.031 | 0.499±0.031 | 0.463±0.019 | 0.518±0.020 | 0.428±0.015 | 0.511±0.038 |
| | 864 | 0.456±0.006 | 0.503±0.008 | 0.468±0.018 | 0.495±0.011 | 0.502±0.008 | 0.552±0.005 | 0.472±0.008 | 0.507±0.005 | 0.537±0.008 | 0.537±0.005 | 0.531±0.007 | 0.531±0.005 | 0.487±0.005 | 0.487±0.004 | 0.500±0.020 | 0.525±0.012 | 0.496±0.014 | 0.546±0.012 | 0.467±0.005 | 0.564±0.005 | 0.467±0.012 | 0.505±0.012 | 0.492±0.024 | 0.531±0.030 | 0.465±0.028 | 0.542±0.034 |
| | 1152 | 0.491±0.006 | 0.521±0.003 | 0.499±0.006 | 0.510±0.006 | 0.523±0.021 | 0.572±0.015 | 0.498±0.011 | 0.521±0.011 | 0.580±0.007 | 0.558±0.004 | 0.599±0.009 | 0.572±0.013 | 0.562±0.029 | 0.533±0.015 | 0.518±0.019 | 0.538±0.019 | 0.511±0.010 | 0.538±0.005 | 0.484±0.002 | 0.578±0.007 | 0.496±0.010 | 0.516±0.008 | 0.523±0.009 | 0.540±0.007 | 0.537±0.056 | 0.602±0.055 |
| $m=0.375$ | 288 | 0.370±0.004 | 0.457±0.002 | 0.383±0.017 | 0.441±0.012 | 0.438±0.016 | 0.499±0.008 | 0.373±0.016 | 0.450±0.009 | 0.436±0.008 | 0.487±0.006 | 0.381±0.008 | 0.451±0.009 | 0.442±0.007 | 0.469±0.004 | 0.451±0.018 | 0.482±0.007 | 0.406±0.015 | 0.479±0.009 | 0.378±0.004 | 0.495±0.004 | 0.384±0.010 | 0.473±0.009 | 0.410±0.020 | 0.491±0.014 | 0.362±0.016 | 0.453±0.012 |
| | 576 | 0.439±0.002 | 0.499±0.002 | 0.467±0.011 | 0.499±0.009 | 0.510±0.009 | 0.561±0.009 | 0.457±0.015 | 0.505±0.006 | 0.521±0.006 | 0.541±0.006 | 0.483±0.006 | 0.512±0.006 | 0.533±0.015 | 0.527±0.012 | 0.522±0.000 | 0.536±0.007 | 0.518±0.021 | 0.568±0.012 | 0.449±0.003 | 0.557±0.006 | 0.470±0.022 | 0.514±0.024 | 0.491±0.031 | 0.544±0.027 | 0.448±0.038 | 0.527±0.039 |
| | 864 | 0.481±0.007 | 0.528±0.005 | 0.497±0.010 | 0.521±0.010 | 0.553±0.022 | 0.598±0.014 | 0.496±0.016 | 0.524±0.011 | 0.580±0.006 | 0.570±0.006 | 0.556±0.006 | 0.555±0.003 | 0.562±0.016 | 0.545±0.010 | 0.562±0.022 | 0.572±0.006 | 0.565±0.017 | 0.604±0.011 | 0.482±0.006 | 0.584±0.004 | 0.494±0.014 | 0.527±0.014 | 0.530±0.022 | 0.561±0.027 | 0.488±0.027 | 0.561±0.028 |
| | 1152 | 0.516±0.008 | 0.547±0.004 | 0.538±0.002 | 0.535±0.002 | 0.575±0.024 | 0.617±0.016 | 0.520±0.002 | 0.540±0.006 | 0.620±0.009 | 0.592±0.005 | 0.626±0.009 | 0.596±0.005 | 0.611±0.016 | 0.568±0.015 | 0.588±0.032 | 0.591±0.020 | 0.573±0.017 | 0.591±0.011 | 0.498±0.001 | 0.598±0.001 | 0.524±0.010 | 0.538±0.010 | 0.553±0.014 | 0.570±0.012 | 0.570±0.058 | 0.626±0.053 |
| $m=0.5$ | 288 | 0.393±0.001 | 0.483±0.001 | 0.422±0.018 | 0.471±0.015 | 0.520±0.029 | 0.558±0.014 | 0.419±0.014 | 0.488±0.004 | 0.511±0.013 | 0.542±0.009 | 0.418±0.006 | 0.484±0.005 | 0.477±0.022 | 0.498±0.008 | 0.561±0.032 | 0.550±0.010 | 0.469±0.022 | 0.526±0.012 | 0.407±0.004 | 0.525±0.004 | 0.405±0.006 | 0.498±0.006 | 0.430±0.006 | 0.516±0.011 | 0.387±0.024 | 0.470±0.018 |
| | 576 | 0.462±0.002 | 0.523±0.002 | 0.517±0.002 | 0.533±0.007 | 0.586±0.026 | 0.621±0.021 | 0.510±0.004 | 0.542±0.007 | 0.587±0.004 | 0.592±0.007 | 0.523±0.005 | 0.545±0.002 | 0.607±0.041 | 0.568±0.030 | 0.623±0.023 | 0.602±0.021 | 0.599±0.027 | 0.624±0.012 | 0.477±0.003 | 0.586±0.002 | 0.498±0.020 | 0.542±0.023 | 0.513±0.044 | 0.571±0.036 | 0.473±0.033 | 0.548±0.054 |
| | 864 | 0.505±0.007 | 0.554±0.005 | 0.551±0.025 | 0.554±0.013 | 0.636±0.035 | 0.661±0.011 | 0.550±0.013 | 0.563±0.005 | 0.645±0.021 | 0.620±0.012 | 0.599±0.005 | 0.589±0.005 | 0.621±0.027 | 0.580±0.013 | 0.662±0.027 | 0.642±0.021 | 0.651±0.026 | 0.665±0.014 | 0.508±0.003 | 0.613±0.003 | 0.533±0.027 | 0.558±0.017 | 0.563±0.019 | 0.596±0.009 | 0.515±0.024 | 0.583±0.028 |
| | 1152 | 0.538±0.007 | 0.572±0.004 | 0.600±0.019 | 0.574±0.006 | 0.654±0.022 | 0.677±0.015 | 0.577±0.012 | 0.582±0.015 | 0.681±0.024 | 0.638±0.007 | 0.673±0.006 | 0.631±0.020 | 0.675±0.043 | 0.609±0.016 | 0.688±0.055 | 0.661±0.029 | 0.653±0.024 | 0.648±0.020 | 0.522±0.001 | 0.625±0.002 | 0.565±0.022 | 0.570±0.017 | 0.583±0.017 | 0.602±0.015 | 0.606±0.068 | 0.649±0.040 |

## USE OF LARGE LANGUAGE MODELS (LLMS) IN PAPER WRITING

LLMs were used *exclusively* for light copy-editing (grammar, clarity, concision) and small LaTeX phrasing/formatting suggestions. We *did not* employ LLMs for retrieval or discovery (e.g., related-work search) or for research ideation. LLMs did not draft technical content or citations and did not contribute at a level comparable to a coauthor. All text and claims were written, checked, and finalized by the authors; any LLM-suggested edits were adopted only after manual verification to prevent hallucinations or unsupported statements.

## ETHICS STATEMENT

This work focuses on methodological contributions for multivariate time-series forecasting. Our experiments rely primarily on publicly available datasets from established academic and institutional sources. In addition, we include one private dataset provided under confidentiality agreements. This dataset contains no personally identifiable information, and we followed all necessary protocols to ensure secure handling, access control, and compliance with data-sharing restrictions. No raw private data will be released; only aggregated statistics and trained model weights are reported in the paper.

Our proposed model, CTF, is designed as a general forecasting framework and does not target specific individuals, groups, or sensitive applications. Potential downstream uses of time-series forecasting models include socially beneficial applications (e.g., energy demand planning, environmental monitoring) as well as sensitive ones (e.g., financial or policy decision-making). We encourage responsible usage, with attention to fairness, transparency, and possible societal impacts. All experiments were conducted under standard computational settings, with environmental costs comparable to typical machine learning research practice.

## REPRODUCIBILITY STATEMENT

We have taken multiple steps to ensure the reproducibility of our work. All model architectures, training procedures, and evaluation protocols are described in detail in the main text and Appendix. Hyperparameters, preprocessing steps, optimization settings, and model configurations are provided in Appendix A.2. For publicly available datasets, we specify their sources in Appendix A.1 to facilitate exact replication. For the private dataset used under confidentiality agreements, raw data cannot be released, but we provide aggregate statistics, evaluation splits, and implementation details sufficient for reproducing the reported results.

Our codebase (including model implementation, training scripts, and evaluation pipeline) will be made available upon publication. All experiments were conducted on standard hardware (NVIDIA RTX 3090 GPUs) with fixed random seeds to reduce variance; each experiment was repeated across three random seeds, and average results are reported. We also provide details on software dependencies (Python version, PyTorch version, and major libraries) to ensure compatibility with common research environments.

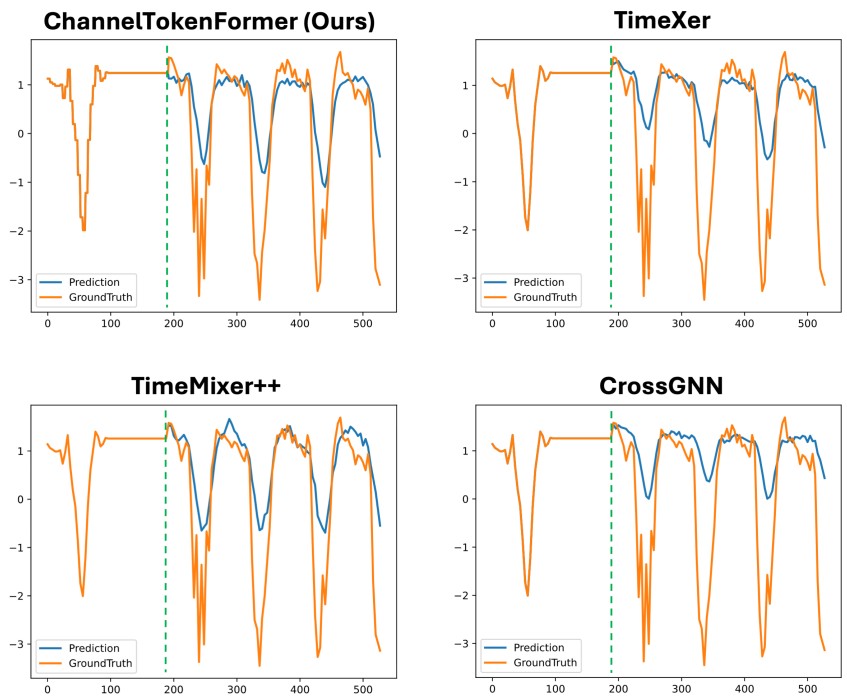

(a) Channel 3 in the ETT1 dataset under asynchronous sampling. Our model accurately captures periodic sharp drops and recoveries, whereas the baselines tend to underpredict their magnitude.

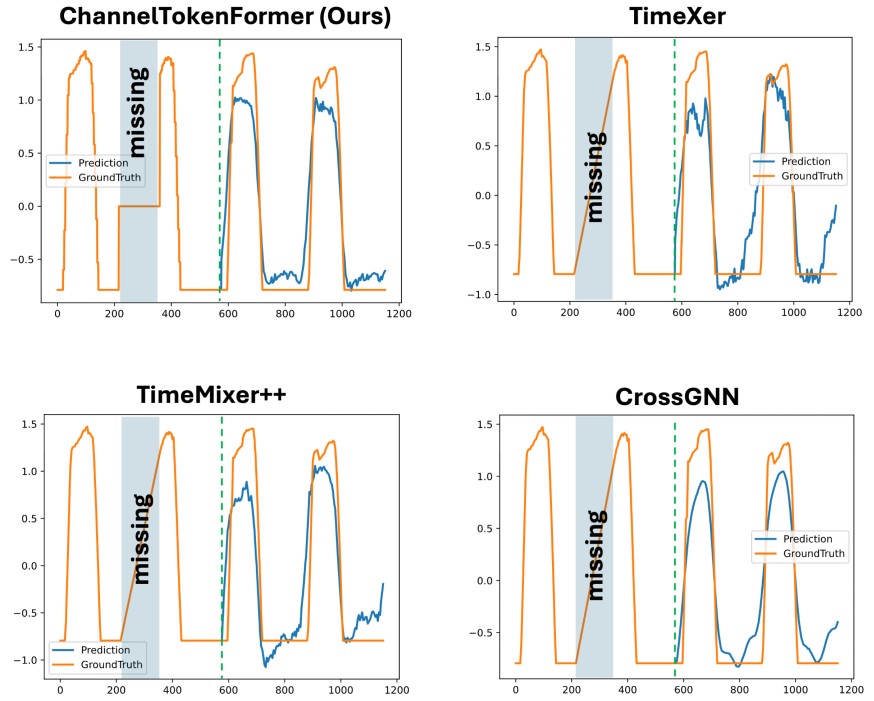

(b) Solar power channel in the SolarWind dataset under asynchronous sampling with block-wise test-time missing. Despite a long missing block, our model robustly captures periodic transitions, while the baselines exhibit noise or trend drift.

Figure 11: Visualization of forecasting results on (a) the ETT1 and (b) the SolarWind datasets.

