# OpenReview forum: "Towards Robust Real-World Multivariate Time Series Forecasting: A Unified Framework for Dependency, Asynchrony, and Missingness"
_ICLR.cc/2026/Conference — ICLR 2026 Poster_

### Official Review · Reviewer_zsWm · 2025-10-23

**Soundness:** 3
**Presentation:** 1
**Contribution:** 2
**Rating:** 4
**Confidence:** 3

**Summary:**

This paper studies the multivariate time series forecasting problem and focus on modeling cross-channel correlations, handling misalignment in sampling frequencies and periods, and dealing with missing data. Based on the challenge that no existing methods properly considered all these three aspects together, this paper proposed a new unified framework for doing that simultaneously. Experiments demonstrate the effectiveness of the proposed framework against existing methods.

**Strengths:**

1. Comprehensive experiments provide evidence that the proposed framework achieves its design goal, and provide insights into the performance behavior of the proposed framework under ablation settings.
2. The proposed framework is straightforward and should be easy to replicate.

**Weaknesses:**

1. While the presentation of the existing challenges is quite detailed and straightforward, the paper could improve on the presentation of the design choices of the proposed method. Right now it is unintuitive how the proposed framework and its core design choices is effective at tackling the aforementioned challenges.
2. The paper could elaborate further on the technical contribution of the proposed method, how it advances on top of existing methods and techniques.

**Questions:**

Could the authors elaborate on the motivation of their proposed framework, so that it is clearer how their proposed framework can effectively tackle the listed challenges and introduce technical contribution?

---

> ### Author Response · Authors · 2025-11-20
>
> We appreciate the reviewer’s comments regarding the clarity of the presentation. We provide below our responses to the two main concerns and one question.
>
> ___
>
> ### **[W1, W2, Q1: Presentation of design choices and contributions]**
>
> In line with the suggestions, we will restructure the manuscript to improve readability by simplifying the problem definition in Section 3 and enhancing the methodological explanation in Section 4 with clear algorithmic descriptions as illustrated below.
>
> ___
>
>  **[Algorithm: Frequency-driven Dynamic Patching and Tokenization]**
>
> **Given**
>
> - Multivariate time series $X \in \mathbb{R}^{N \times L}$ with $N$ channels and input length $L$
> - Sampling periods $\{s_i\}_{i=1}^N$ for each channel $i$
> - Dominant periods $\{D_i\}_{i=1}^N$ for each channel $i$
> - Prediction horizon $H$
> - Masking probability $q$
> - Number of channel tokens $m$
>
> ---
>
> I. Compute the minimum sampling period:
>    - $s_{\min} = \min_i s_i$
>
> II. For each channel $i = 1, \dots, N$:
>
>    1. Compute the relative sampling period and the number of valid time points:
>       - $s_i^{\prime} = s_i / s_{\min}$
>       - $L_i = L / s_i^{\prime}$
>
>    2. Define the patch length and the number of patches:
>       - $p_i = D_i / s_i^{\prime}$  (patch length in samples)
>       - $K_i = L_i / p_i = L / D_i$  (number of patches for channel $i$)
>
>    3. Consider the univariate series $X^{(i)} \in \mathbb{R}^{L_i}$ and partition it into patches
>       $P_1^{(i)}, \dots, P_{K\_i}^{(i)}$, where each patch
>       $P^{(i)}_j \in \mathbb{R}^{p_i}$.
>
>    4. Initialize the list of local tokens for channel $i$:
>       - $\mathcal{T}^{(i)} \leftarrow [\ ]$
>
>    5. For each patch $j = 1, \dots, K_i$:
>       - If the patch $P^{(i)}_j$ is fully unobserved, or randomly masked
>         with probability $q$ (i.e., $\mathrm{Bernoulli}(q) = 1$),
>         then **skip** this patch.
>       - Otherwise, compute the local token with positional embedding
>         $$T_j^{(i)} = W_{p_i} P_j^{(i)} + e_{i,j}^{\mathrm{pos}} \quad \text{where } W_{p_i} \in \mathbb{R}^{d \times p_i},\text{ } T_j^{(i)}
>         \in \mathbb{R}^d.$$
>       - Append $T^{(i)}_j$ to the list $\mathcal{T}^{(i)}$.
>
>    6. Initialize $m$ learnable channel tokens for channel $i$:
>       - $C^{(i)}_1, \dots, C^{(i)}_m \in \mathbb{R}^d$
>
>    7. Construct the token sequence for channel $i$ by concatenation and adding the channel embedding:
>       $$
>       S_i^{(i)} = \big[ T_1^{(i)}, \dots, T_{K_i}^{(i)}\; C_1^{(i)}, \dots, C_m^{(i)} \big] + e_{i}^{\mathrm{chn}}.
>       $$
>
>       where $e_{i}^{\mathrm{chn}} \in \mathbb{R}^d$ is broadcast and added to every token in $S_i^{(i)}$.
>
>
> III. Concatenate all channel-wise sequences to obtain the final token sequence
>    $
>    S = \big[ S^{(1)}, \dots, S^{(N)} \big].
>    $
>
> IV. Return $S$ as the input token sequence for the subsequent Transformer encoder.

---

> ### Author Response · Authors · 2025-11-20
> **Official Comment by Authors (Cont')**
>
> We next provide a more precise explanation of the design choices behind the proposed framework, including its motivation and how it addresses the three challenges.
>
> ---
> ### Our Contribution
>
> Our work addresses a critical yet underexplored gap in time series forecasting: **the simultaneous handling of three core real-world challenges.**
>
> - **Challenge A - Channel-wise asynchronous sampling**: Misaligned input sequences resulting from distinct sampling periods for each channel.
> - **Challenge B - Test-time missing blocks**: Contiguous missing intervals at test time, commonly arising from sensor failures.
> - **Challenge C - Inter-channel dependencies**: Complex and often noisy correlations across channels that are difficult to model reliably.
>
> While prior works have addressed these individually, no existing approach robustly tackles all three together.
>
> Specifically:
> - **Channel-dependent methods** focus on challenge C **but are not originally designed to handle A and B.**
>   - TimeXer and iTransformer, with some modifications, can manage A and C **but still struggle with B.**
> - **Channel-independent methods** might be an alternative for A, **but they fundamentally overlook C.**
> - **Specific approaches for missing values** address B and C **but still do not handle A.**
> - **Methods for irregular time series are not suited to robustly handle A and B** especially over long lookback and horizon settings.
>   - They primarily focus on non-uniform sampling within each channel and high sparsity.
>
> Our work integrates targeted strategies for A, B, and C into a robust, unified framework designed to accommodate their complex interplay.
>
> The key components we adopt and adapt include:
> - **Frequency-aware dynamic patching**: Supports non-aligned, variable-length patching **to handle A effectively.**
> - **Random patch masking**: Explicitly trains the model **to enhance robustness to B.**
> - **Channel tokens as attention anchors**: **Crucial bridging component that summarizes variable-length patches (A), enables robust decoding under missing blocks (B), and facilitates inter-channel attention (C).**
> - **Attention masking over read-only tokens**: Prevents local patterns from being diluted by global summaries.
>
> **Again, our contribution lies in the integration of these elements into a unified and robust framework specifically designed to accommodate the complex interplay of three core real-world challenges.**

---

> ### Comment · Reviewer_zsWm · 2025-11-25
>
> I would like to thank the authors for their detailed response, which largely clarified my doubts raised in my original assessment.
> I also appreciate that the authors have revised their paper, which has improved the presentation of their work.
> I have raised my rating accordingly.

---

> > ### Author Response · Authors · 2025-11-25
> >
> > We sincerely thank the reviewer for taking the time to carefully re-examine our manuscript in order to clarify the points that were previously unclear.
> > We are truly grateful for your overall positive assessment of our work and for the improved rating, which we take as strong encouragement to further refine and advance this line of research.

---

### Official Review · Reviewer_e4Zq · 2025-10-27

**Soundness:** 3
**Presentation:** 3
**Contribution:** 3
**Rating:** 6
**Confidence:** 4

**Summary:**

This paper introduces ChannelTokenFormer (CTF), a novel Transformer-based framework for multivariate time series forecasting designed to be robust to real-world data challenges. The authors identify three key challenges that often co-occur in practice but are typically addressed in isolation by existing methods: (1) complex inter-channel dependencies, (2) channel-wise asynchronous sampling (different channels having different sampling rates), and (3) block-wise missing values at test time.

The core contribution is a unified model that handles these three challenges simultaneously without relying on interpolation, which can introduce spectral distortion. The key components of CTF are: 1.Channel Tokens: Repurposed from prior work, these tokens act as global summaries or "attention anchors" for each channel. 2. Frequency-Based Dynamic Patching: Each channel is patched non-uniformly based on its dominant frequency (detected via FFT) and sampling rate, naturally accommodating asynchrony. 3. Unified Mask-Guided Attention: A single attention mechanism processes both local (patch) tokens and global (channel) tokens. A carefully designed attention mask controls information flow, allowing local tokens to attend only within their channel, while channel tokens can aggregate information from their own local tokens and other channel tokens, enabling cross-channel communication. 4. Patch Masking: Missing blocks are handled by simply removing the corresponding local tokens from the input, a strategy for which the model is prepared via random patch masking during training.

The authors conduct extensive experiments on "practical" versions of four standard benchmarks (ETT, Weather, SolarWind) and two real-world datasets (EPA-Air, LNG Cargo Handling). The results demonstrate that CTF consistently outperforms a wide range of state-of-the-art models in settings with asynchronous sampling and test-time missingness.

**Strengths:**

1. The paper tackles a crucial, real-world problem by addressing channel dependency, asynchrony, and missingness in a unified manner, moving beyond the idealized assumptions common in much of the literature.
2. The proposed ChannelTokenFormer, with its unified mask-guided attention, offers an elegant solution that avoids signal-distorting interpolation. The integration of frequency-based dynamic patching is a smart way to handle heterogeneous sampling rates.
3. The paper is exceptionally clear, well-written, and illustrated, making the contributions easy to understand and appreciate. The detailed appendix further underscores the quality and care put into the research.

**Weaknesses:**

1. The authors compare their method with mainly channel-dependent transformer-based methods. However, many methods have achieved SOTA performance with non-transformer architectures [1] [2]. Comparison with these methods is necessary for a comprehensive evaluation.

2. The section introducing the research methodology lacks essential mathematical formulas and specific descriptive details, which creates certain difficulties for readers to fully understand the implementation logic and operational steps of the proposed method. It is recommended to supplement the core formulas and elaborate on key technical parameters or procedural details to enhance the clarity and reproducibility of the methodology.

3. The unified attention mechanism has a computational complexity that is quadratic in the total number of tokens (patches + channel tokens). The authors' own analysis shows OOM at 280 channels on a 24GB GPU. While sufficient for many applications, this could be a limitation for very high-dimensional problems with thousands of channels, such as the Traffic prediction problems [3]

4. This article uses frequency-domain information for assistance, but it does not conduct a comparison of frequency-domain methods [4][5].

[1]  Si-An Chen, Chun-Liang Li, Sercan Ö. Arik, Nathanael C. Yoder, Tomas Pfister: TSMixer: An All-MLP Architecture for Time Series Forecast-ing. Trans. Mach. Learn. Res. 2023 (2023)
[2] Han Lu, Xu-Yang Chen, Han-Jia Ye, De-Chuan Zhan: SOFTS: Efficient Multivariate Time Series Forecasting with Series-Core Fusion. NeurIPS 2024
[3] http://pems.dot.ca.gov/
[4] Zhijian Xu, Ailing Zeng, Qiang Xu: FITS: Modeling Time Series with 10k Parameters. ICLR 2024
[5] Kun Yi, Qi Zhang, Wei Fan, Shoujin Wang, Pengyang Wang, Hui He, Ning An, Defu Lian, Longbing Cao, Zhendong Niu: Frequency-domain MLPs are More Effective Learners in Time Series Forecasting. NeurIPS 2023

**Questions:**

1. The paper mentions the FREQUENCY BIAS. How does the bias relate to the forecasting performance? Can it be alleviated by a more specific design in the frequency domain?

---

> ### Author Response · Authors · 2025-11-20
>
> We appreciate your valuable comments and suggestions on our architectures and experiments. There are four main concerns and one question regarding our methods and experiments. We provide corresponding explanations and additional experiments for each point in the following order.
>
> ---
> ### **[W1: Comparison with non-transformer baselines]**
>
> We have already compared our method with strong non-Transformer approaches, including TimeMixer++ and various graph-based models such as CrossGNN and tPatchGNN. Additionally, as suggested in the reviews (including the review from Reviewer *9xPZ*), we conducted further comparisons with models that incorporate clustering modules (e.g., DUET) and frequency-based methods such as FreTS and FITS. We believe that these experiments constitute a sufficiently comprehensive evaluation against non-Transformer techniques.
>
> ---
> ### **[W2: Readability issues & Clarification on hyperparameter specification]**
>
> We appreciate the reviewers’ feedback (including the review from Reviewer *657H*) regarding the complexity and clarity of Sections 3 and 4, and we will reorganize these sections in the revised version as follows:
>
> - In Section 3, we will **simplify the problem formulation to more intuitively present the concepts of asynchronicity and test-time block missingness.** Detailed explanations related to the proposed method will be moved to Section 4.
> - In Section 4, we will describe the dynamic patching strategy and the channel token mechanism **in an algorithmic format.** Based on this, we will provide a more detailed explanation of the implementation and operational procedure of our proposed methodology to improve readability. We will also revise and **disambiguate overlapping notations and supplement the core mathematical formulations where necessary.** The algorithm itself can be found in our response to reviewer *zsWM*.
>
> The implementation details are provided in Appendix A.2, and **the accompanying code explicitly specifies all technical parameters required for reproducibility.**
>
> ---
> ### **[W3: Concern about computational complexity]**
>
> We agree that the unified attention mechanism has quadratic complexity in the total number of tokens and therefore incurs nontrivial memory overhead as the number of channels grows.
> The increased complexity mainly comes from our design choice to have the model simultaneously capture fine-grained dynamics within each channel and pairwise interactions across all channels.
>
> At the same time, this does not translate into a prohibitive slowdown for realistic channel counts. As noted in Appendix C.3, on a dataset with 200 channels the measured inference latency is 0.917s for CTF versus 0.903s for TimeXer, indicating that our method remains practically usable for datasets with a moderate number of channels, which covers many real-world applications.
>
> For high-dimensional regimes with thousands of channels like Traffic, **additional mechanisms such as explicit channel sparsification or grouped channel representations would be needed to further improve memory efficiency, and we regard these extensions as important future work.**

---

> ### Author Response · Authors · 2025-11-20
> **Official Comment by Authors (Cont')**
>
> ---
> ### **[W4: Comparison with frequency-domain methods]**
>
> We appreciate the suggestion. Comparing our approach with frequency-domain baselines is indeed a valuable way to highlight the strengths of our model. We have compared our method with the suggested FreTS and FITS models. The average CMSE results across four prediction lengths, following the experimental setup described in the paper, are provided in the table below.
>
> | Dataset | CTF  | FreTS    | FITS  |
> |--------|:-------:|:-------:|:-------:|
> | ETT1   | **0.399** |  0.436 | 0.443 |
> | ETT2   |  **0.377** | 0.440 | 0.391 |
> | SW     |  **0.403** | 0.416 | 0.485 |
> | Weather|  **0.275** | 0.285 | 0.297 |
> | EPA    |  **0.776** | 1.018 | 1.245 |
> | CHS    | **0.285** | 0.485 | 0.289 |
>
> Since our baseline performs frequency-driven patching tailored to each channel and adopts a Transformer-based architecture, it consistently achieves superior performance. In contrast, the two models exhibit significant degradation, particularly on real-world datasets with strong channel-wise asynchronous sampling such as EPA and CHS. The experimental results of these two models will also be included in the revised paper.
>
> ---
> ### **[Q1: Frequency bias and its impact on forecasting performance]**
> As shown in our spectral analysis in Appendix B, interpolation at the input stage already acts as an undesirable low-pass filter, distorting the spectrum before the forecasting model even operates on the data.
>
> In our view, this input-level distortion is a primary source of frequency bias and its impact on forecasting performance.
> For this reason, we believe that **the first and most critical step is not to add a more elaborate frequency-domain head, but to adopt an interpolation-free architecture that can ingest asynchronously sampled, partially missing multivariate series without forcing them onto an artificially regular grid.**
> Our CTF is designed precisely with this goal in mind.
>
> More specialized modules operating explicitly in the frequency domain could be explored on top of such an interpolation-free backbone, but a detailed design and evaluation of those components is beyond the scope of this work and is left for future research.

---

### Official Review · Reviewer_9xPZ · 2025-10-28

**Soundness:** 2
**Presentation:** 2
**Contribution:** 2
**Rating:** 2
**Confidence:** 5

**Summary:**

A practical forecasting setting is introduced where three core real-world challenges occur simultaneously: channel-wise asynchronous sampling, block-wise missing intervals, and complex crosschannel dependencies. Instead of interpolation, this approach handles both missing segments and sampling gaps through masking and frequency-based dynamic patching.

**Strengths:**

1、Multivariate time series forecasting is important to various domains.

2、There are quite a few nice illustrations.

3、This work focuses on an important problem that could have real-world applications.

4、The figures and tables used in this work are clear and easy to read.

**Weaknesses:**

1、The proposed algorithm has notable limitations. The authors should further clarify whether their method can effectively handle missing values during training, especially in scenarios where missing values appear in a discrete rather than continuous manner. It remains unclear how the model ensures robustness and applicability under such conditions.

2、In addition, although the authors claim that their method addresses three key challenges in real-world scenarios—variable modeling, multi-source asynchrony, and missing values—many other critical issues in time series analysis are not discussed, such as non-stationarity, concept drift, and probabilistic forecasting. As a result, the paper’s overall logic appears somewhat fragmented, lacking a coherent and systematic research focus.

3、In the comparison section, the authors only include several Channel-Dependent methods from 2023 and 2024. However, as a paper submitted to ICLR 2026, it fails to compare against more recent representative works published in 2025, which weakens the experimental credibility and timeliness.

4、In lines 295–296, the authors claim that their method ensures computational efficiency. However, from an implementation perspective, introducing a mask mechanism does not actually reduce computational costs. On the contrary, when the dataset contains a large number of variables, the computational overhead of the proposed method may increase significantly, potentially reducing overall efficiency.

5、In the NIPS 2024 workshop[1], some researchers pointed out that current methods sometimes use the "drop-last" trick [2] to improve performance. Therefore, It is recommended that you clarify whether the "drop - last" operation was used in your paper in the implementation details section of your paper for transparency.

If my problem is solved, I am willing to improve my score.

[1] Fundamental limitations of foundational forecasting models: The need for multimodality and rigorous evaluation

[2] TFB: Towards Comprehensive and Fair Benchmarking of Time Series Forecasting Methods

**Questions:**

1、The proposed algorithm appears to have notable limitations. Can the authors clarify whether their method can effectively handle missing values during training, especially when the missing values occur in a discrete rather than continuous manner? How does the model ensure robustness and applicability under such conditions?

2、Furthermore, while the authors claim that their method addresses variable modeling, multi-source asynchrony, and missing values, have they considered other equally important challenges in time series analysis, such as non-stationarity, concept drift, and probabilistic forecasting? Could the omission of these aspects make the overall logic of the paper appear fragmented and lacking a coherent research focus?

3、In the comparison section, the authors only include Channel-Dependent methods from 2023 and 2024. Why are more recent representative works from 2025, such as **TimeFilter** and **DUET**, not included in the comparison? Would the absence of these up-to-date baselines weaken the credibility and timeliness of the experimental results?

4、Lastly, in lines 295–296, the authors claim that their method ensures computational efficiency. However, does introducing a mask mechanism actually reduce computational costs? When the dataset contains a large number of variables, wouldn’t the computational overhead instead increase significantly, thereby affecting the overall efficiency?

If my problem is solved, I am willing to improve my score.

---

> ### Author Response · Authors · 2025-11-20
>
> We appreciate your valuable comments and suggestions on our architectures and experiments. There are five main concerns and five questions regarding our methods and experiments. We provide corresponding explanations and additional experiments for each point in the following order.
>
> ---
> ### **[W1, Q1: Concern about handling discrete missing]**
>
> We respectfully disagree with the implication that our method cannot handle discrete missing values or that this would constitute an unaddressed limitation.
>
> First, the real-world benchmarks we use **already contain discrete missingness**.
> For example, in ETTm (15-minutely) data, one can observe that missing entries are implicitly encoded as zeros.
>
> Not only our model but also other baselines rely on **a small neural tokenizer (patch embedding network)** to process these inputs.
> We empirically observe that this stage **already provides a reasonable degree of robustness to discrete missing values.**
>
> **Thus, we are not ignoring or avoiding discrete missingness; rather, it is already handled to a practical extent by the standard modeling pipeline shared across methods.**
> In addition, we have **already conducted experiments with random, scattered missing values** ranging from 5 to 20, which are much shorter than the patch lengths.
> The experimental results are reported in Tables 19 and 20.
>
> **Our main contribution is to go beyond this default robustness and explicitly tackle more challenging and realistic patterns, including naturally occurring block-wise missing intervals** caused by short-term sensor malfunctions or logging failures.
>
> ---
> ### **[W2, Q2: Scope of challenges considered]**
>
> We would like to clarify the intended scope of this work. Our primary goal is to address, in a unified manner, three challenges that are both widely recognized and heavily studied in time series forecasting: (i) channel-dependent modeling, (ii) channel-wise asynchrony, and (iii) block-wise missingness. This focus is **explicitly stated in the title and introduction**, and our architectural and experimental designs are organized around these three aspects.
>
> We fully acknowledge that other important challenges in time series forecasting exist, such as concept drift. However, we consider these aspects to be beyond the primary scope of this work, rather than additional objectives that must be solved within a single paper.
>
> In fact, non-stationarity is already a property of many benchmark datasets we use, so our experiments inherently evaluate CTF and other baselines under non-stationary conditions.
> Even though we follow the standard deterministic “direct forecasting” protocol adopted in prior work, probabilistic forecasting could also be incorporated on top of CTF, for example by sampling multiple prediction trajectories or by equipping the model with a suitable probabilistic output head.
> However, a full probabilistic treatment is beyond the scope of the current study.
>
> Concept drift and test-time adaptation are likewise meaningful directions, particularly in settings where sensor data regimes change over time. We agree that extending CTF with mechanisms for drift-aware or test-time adaptive forecasting would be valuable, and we regard this as an interesting avenue for future work.
>
> **To make this scope explicit, we will revise the introduction to briefly acknowledge other important challenges in time series forecasting, but then clearly state that this paper deliberately concentrates on jointly addressing the above three challenges, which are among the most widely studied and practically impactful.**

---

> ### Author Response · Authors · 2025-11-20
> **Official Comment by Authors (Cont')**
>
> ---
> ### **[W3, Q3: Comparison with recent channel-dependent baselines]**
>
> We appreciate the reviewer’s suggestion regarding the inclusion of more recent works from 2025. Notably, Hi-Patch [1] is also a 2025 channel-dependent approach targeting irregular time-series forecasting. Although TimeFilter and DUET were not initially considered due to the lack of direct comparison with our baseline model TimeXer [2], we have subsequently conducted additional experiments. The average CMSE results across four prediction lengths, following the experimental setup described in the paper, are provided in the table below.
>
> | Dataset |   CTF   | TimeFilter |  DUET  |
> |-------|:-------:|:----------:|:------:|
> |  ETT1   |  **0.399**  |   0.412    | 0.424  |
> |  ETT2   |  **0.377**  |   0.383    | 0.388  |
> |   SW    |  **0.403**  |   0.404    | 0.438  |
> | Weather |  0.275  |   **0.273**    | 0.309  |
> |   EPA   |  **0.776**  |   0.863    | 0.782  |
> |   CHS   |  **0.285**  |   0.304    | 0.307  |
>
> The results show that our method achieves competitive baseline performance relative to these models. We acknowledge that incorporating recent non-Transformer approaches is valuable for completeness. Therefore, in the revised version, we will include these recent methods as additional channel-dependent baselines to enhance the timeliness and comprehensiveness of our experimental evaluation.
>
> [1] Hi-Patch: Hierarchical Patch GNN for Irregular Multivariate Time Series. ICML’25.
>
> [2] TimeXer: Empowering Transformers for Time Series Forecasting with Exogenous Variables. NeurIPS’25.
>
> ---
> ### **[W4, Q4: Concern about computational complexity]**
>
> We apologize for the confusion caused by our wording.
>
> In the sentence *“the masking strategy ensures computation remains efficient by restricting attention to necessary token interactions,”* **our intention was to describe efficiency in terms of information flow, specifically by avoiding unnecessary token interactions, rather than to claim a reduction in the asymptotic computational complexity of attention.**
>
> We are fully aware of this computational complexity as a limitation and explicitly acknowledge it in the conclusion, and we agree that the unified attention mechanism has quadratic complexity in the total number of tokens, thereby incurring nontrivial memory overhead as the number of channels grows.
> The increased complexity mainly comes from our design choice to have the model simultaneously capture fine-grained dynamics within each channel and pairwise interactions across all channels.
>
> At the same time, this does not translate into a prohibitive slowdown for realistic channel counts. As noted in Appendix C.3, on a dataset with 200 channels the measured inference latency is 0.917s for CTF versus 0.903s for TimeXer, indicating that our method remains practically usable for datasets with a moderate number of channels, which covers many real-world applications.
>
> **For high-dimensional regimes with thousands of channels like Traffic, additional mechanisms such as explicit channel sparsification or grouped channel representations would be needed to further improve memory efficiency, and we regard these extensions as important future work.**
>
> ---
> ### **[W5: Usage of drop-last trick]**
>
> We appreciate the reviewer’s suggestion regarding the reliability of the experiments. **We did not apply the drop-last operation**, and we will explicitly clarify this in the Appendix A.2 ("Implementation Details").

---

> > ### Comment · Reviewer_9xPZ · 2025-11-21
> > **Thank you for your detailed response！**
> >
> > Thank you for your detailed response. Since ICLR allows revisions to the submitted paper, I suggest incorporating the relevant changes directly into the manuscript. These modifications will significantly improve the overall quality and clarity of the paper. After reviewing your updated version, I will accordingly adjust my score.

---

> ### Author Response · Authors · 2025-11-23
> **Revisions Implemented in Response to Reviewer Feedback**
>
> We thank the reviewer for the helpful suggestions regarding revisions to the manuscript.
> Below, we summarize the main changes, incorporating suggestions from the reviewer as well as others, which are highlighted in red in the revised submission.
>
> ---
> - **Clarification of the Scope of Challenges (Introduction)**
>   - In the opening paragraph, we explicitly acknowledge that real-world time series forecasting faces a wide range of practical challenges beyond those addressed in our work.
>   - We then clarify why our study focuses on three key challenges: inter-channel dependencies, channel-wise asynchrony, and block-wise missingness, which are often oversimplified in conventional multivariate settings.
>
> - **Discrete Missingness (Introduction & Appendix C.6)**
>   - Before discussing long contiguous missing intervals, we now briefly mention discrete missing values and explain that such patterns already exist in standard benchmarks (e.g., ETTm).
>   - A detailed analysis of discrete missingness, based on empirical evidence from the embedding visualization in Appendix C.6 together with the accompanying discussion, shows that our small patch embedding network already exhibits a certain degree of inherent robustness to scattered missing values.
>   - Specifically, Appendix C.6 provides a discussion of handling discrete missing values, including an explanation of why the small patch embedder is robust to discrete missingness, framed through smooth (approximately Lipschitz) neural mappings, and additional experimental results under discrete missingness on existing benchmarks.
>
>   Additionally, we revised part of Case 2 in Section 5 (Experiments) to emphasize that the experiments evaluate robustness under block-wise missing intervals rather than mere discrete missingness (*"beyond discrete missing values"*).
>
> - **Expanded Related Work & Experiments on Channel-Dependent Approaches (Section 2 & 5)**
>   - We updated the related work on channel-dependent strategies to include recent advances that involve sparse/clustered inter-channel structures (e.g., DUET) as well as TimeFilter, which we additionally evaluate in the experiments.
>   - We added experimental comparisons with the latest 2025 channel-dependent baselines, TimeFilter and DUET.
>   - Even under realistic conditions in which all three challenges coexist, CTF continues to achieve the best or near-best performance on most benchmark datasets.
>
> - **Clarification on Computational Complexity and Limitations (Section 4 & Conclusion)**
>   - To avoid misunderstandings, we revised the final paragraph of Section 4 and the Conclusion to more explicitly state the limitations of our approach.
>   - In particular, while CTF is practical for a wide range of real-world channel configurations, ultra–high-dimensional regimes will require additional mechanisms (e.g., channel sparsification or grouping), which we outline as future work.
>
> - **Dataset Validity and Additional Details (Appendix A.1, A.2)**
>   - Following another reviewer’s concern, Appendix A.1 now provides further justification for the dataset configurations used in our experiments.
>   - Appendix A.2 clearly states that detailed hyperparameter settings are fully available in the accompanying code, and **clarifies that we do not use tricks such as “drop-last.”**
>   - We also added a more explicit description of how missing blocks are injected during evaluation.
>
> - **Additional Experiments on Frequency-based Approaches (Section 5 & Appendix C.5)**
>   - Following another reviewer’s suggestion, we also incorporated comparisons with frequency-based methods (FreTS, FITS) given that CTF employs frequency-domain information for dynamic patching.
>   - Results, included in Appendix C.5, show that interpolation-free modeling prevents spectral distortion under channel-wise asynchrony, leading to substantially better performance than frequency-based baselines.
>
> - **Frequency Bias & Implications for Forecasting Performance (Appendix B.3)**
>   - Following another reviewer’s suggestion, we highlight that input-level distortions induce frequency bias that degrades forecasting performance, motivating interpolation-free architectures rather than more complex frequency-domain heads on interpolated inputs.
>   - We substantiate this claim through additional experiments on frequency-based approaches and a frequency-bias analysis of predicted series.
>
> In addition, we are actively working to further improve *the readability of Sections 3 (Problem) and 4 (Method)*.
>
> We hope that these revisions address your concerns and will be viewed positively.

---

> > ### Comment · Reviewer_9xPZ · 2025-11-26
> > **Thank you for your detailed reply. Your response has effectively resolved my issue.**
> >
> > Thank you for your detailed reply. Your response has effectively resolved my issue. I have decided to revise my score to support the acceptance of this paper. Good luck！

---

> > > ### Author Response · Authors · 2025-11-26
> > >
> > > Thank you for reconsidering our work and for the revised score in support of acceptance.
> > > We are very grateful that our response has resolved your concerns!

---

### Official Review · Reviewer_657H · 2025-10-28

**Soundness:** 3
**Presentation:** 2
**Contribution:** 2
**Rating:** 4
**Confidence:** 4

**Summary:**

This paper introduces ChannelTokenFormer, a unified Transformer-based framework targeting robust multivariate time series forecasting under three often neglected but crucial real-world conditions: channel-wise asynchronous sampling, block-wise missingness at test time, and complex cross-channel dependencies. The model employs a mask-guided unified attention mechanism with channel tokens, frequency-driven dynamic patching per channel, and explicit patch masking for missingness during training and inference. The approach is evaluated on several benchmark and real-world datasets (including channel-heterogeneous resamplings and a large-scale industrial application), consistently demonstrating improved performance versus state-of-the-art baselines, both under idealized and challenging practical setups.

**Strengths:**

1. The paper formalizes a comprehensive and realistic forecasting challenge: simultaneous handling of dependency, asynchrony, and missingness, as illustrated clearly in Figure 1, moving beyond isolated treatment of these aspects in prior works.
2. The paper introduces a unified mask-guided attention mechanism for channel tokens (see Figure 2 and Figure 3) that elegantly separates local intra-channel modeling from cross-channel global aggregation, supporting asynchronous and missing inputs naturally.
3. The model integrates frequency analysis (FFT-driven patch length selection) to respect per-channel periodicities and sampling densities, supported by theorized and empirical evidence (Appendix, Table 9).
4.  Patch masking for both training and test scenarios is well-motivated and shown to bolster model robustness. The ablation studies reinforce the necessity and effectiveness of each mechanism, especially for structured block-wise missingness.
5.  Detailed quantitative experiments across a rich collection of public and industrial datasets, as summarized in Tables 1 and 2, show state-of-the-art performance. Ablations in the appendix clarify the interplay among model components.
6.  The paper provides in-depth analysis of interpolation-induced spectral distortion justifying the benefits of the interpolation-free approach and thoughtfully connecting theory with practical modeling.
7. The method's scalability to high channel counts and long sequences is empirically demonstrated (**Table 12, Table 13**), supporting claims of practical deployability.

**Weaknesses:**

1.  While practical motivation for the mask-guided attention and frequency-based patching is strong, the formal theoretical analysis is somewhat lacking.  For example, the impact of frequency-based patching on generalization is empirically supported but not mathematically justified.
2. The bulk of the validation is empirical, with most arguments for effectiveness given by ablations and performance gains. The absence of principled theoretical results or proofs (e.g., why the particular masking scheme is optimal or robust under certain classes of missingness) is noticeable. For example, the construction of attention masks (as shown in **Figures 7, 8, 9**) is justified via heuristic motivation rather than analytical optimality or formal properties.
3. The manuscript provides a detailed exposition of the proposed method; however, certain sections are densely structured and may be difficult for readers to follow. In addition, the notation conventions—such as the definitions of the mask matrix, channel tokens, and sampling factors (see Sections 3 and 4)—are somewhat cumbersome and could benefit from simplification or clearer explanations. Moreover, some formulas are embedded directly within the main text, which further hinders readability and could be better presented as numbered equations or in a dedicated formula environment.
4. The modification of canonical datasets into "practical" versions (e.g., Weather-practical, SolarWind) relies on synthetic resampling and missingness injection, which may bias the comparison to baselines. While the intention and utility of these datasets are strong, it is unclear how they reflect real-world statistical complexity versus controlled synthetic heterogeneity, and possible overfitting to constructed artifacts cannot be ruled out.
5.  While the main architecture is described at a reasonably high level, some key hyperparameters are not specified in detail and are only provided as approximate ranges. Given the relatively complex pipeline and variable-dependent model configuration, this may hinder independent replication.
6. Appendix B discusses the spectral effects of interpolation (with supporting figures), but the argument remains somewhat qualitative. Providing a more comprehensive, formal, and comparative mathematical analysis, including its impact on prediction accuracy and data outcomes, could enhance the scientific value and make the frequency-related discussion more broadly applicable.
7.  The paper acknowledges some scalability and generalization challenges in the conclusion but otherwise does not engage deeply with possible failure modes or cases where ChannelTokenFormer might underperform (e.g., non-stationary environments, highly irregular or bursty missingness patterns, or settings with weak cross-channel structure).
8. Limited Superiority in Conventional Settings: Although the model is evaluated on regularly sampled multivariate time series without missing values to demonstrate its general applicability, the results shown in Table 15 indicate that it does not consistently outperform existing state-of-the-art baselines in this conventional forecasting setting. This suggests that the architectural advantages of ChannelTokenFormer—designed primarily for handling asynchrony and missingness—may not translate into clear gains when data are fully observed and regularly sampled.

**Questions:**

1. Can the authors provide more insights or theoretical grounding for why the proposed mask-guided attention scheme is robust to a wide variety of missingness patterns and channel asynchrony, beyond heuristic and experimental support?
2. How sensitive is the model to hyperparameters governing patch size selection—especially when the dominant period is weak or ambiguous? Could a poorly tuned FFT threshold degrade performance?
3. Regarding dataset construction: can the authors clarify to what extent the practical benchmark setups (modified from canonical datasets) reflect organic missingness and sampling asynchrony found in the wild, rather than synthetically controlled structures? Can the robustness of CTF be validated on additional non-synthetic real-world datasets?
4. Could the authors expand on which failure modes arise with their approach (e.g., with highly bursty, uncorrelated missingness, or in scenarios with low cross-channel redundancy)?
5. As shown in Table 15, the model does not achieve state-of-the-art performance on long-horizon forecasting tasks under conventional multivariate settings. Does this indicate that its practical effectiveness may be limited when the data are fully observed and regularly sampled?

**Details Of Ethics Concerns:**

No.

---

> ### Author Response · Authors · 2025-11-20
>
> We appreciate your valuable comments and suggestions on our architectures and experiments. There are eight main concerns and five questions regarding our methods and experiments. We provide corresponding explanations for each point in the following order.
>
> ---
>
> ### **[W1, W2, Q1: Suggestion for theoretical background of our modeling]**
>
> We agree that having an explicit theoretical foundation would further strengthen our work.
>
> However, a complete mathematical theory of robustness in such a complex setting, would inevitably rely on strong assumptions, and also **we believe this is somewhat orthogonal to the empirical and modeling approach we pursue in this work.**
>
> As the reviewer already notes, the practical motivation behind our modeling choices is reasonable. Here, we clarify that our model’s components are not ad-hoc heuristics, but are grounded in explicit architectural inductive biases about how information should flow between local and channel-level representations under **real-world multivariate conditions: channel-wise asynchronous sampling (A), test-time missing blocks (B), and inter-channel dependencies (C).**
>
> Our work integrates targeted strategies for A, B, and C into **a robust, unified framework** designed to accommodate their complex interplay.
>
> - **Frequency-aware dynamic patching**
>   - Supports non-aligned, variable-length patching **to handle A effectively.**
>   - Designed with the philosophy that patches should correspond to minimal semantic units, analogous to words.
>   - Already sufficiently validated in prior works [3, 4], which we also cite in our paper.
>
> - **Random patch masking**
>   - Explicitly trains the model **to enhance robustness to B.**
>
> - **Channel tokens as attention anchors**
>   - **Serve as a crucial bridging component that summarizes variable-length patches (A), enables robust decoding under missing blocks (B), and facilitates inter-channel attention (C).**
>   - Encourage the model to capture high-level, relatively stable inter-channel dependencies rather than overfitting to noisy or transient local relationships.
>   - Models that directly couple fine-grained local tokens across channels (e.g., Crossformer [1]) can be sensitive to noise in local relationships.
>   - Conceptually aligned with architectures that operate on per-channel summaries, such as iTransformer [2], which also emphasize stable, high-level interactions through summarized tokens.
>
> - **Attention masking over read-only tokens**
>   - Prevents local patterns from being diluted by global summaries.
>
> We therefore believe that, even in the absence of a theory, these design choices are functionally well justified for the realistic multivariate settings. A theoretical treatment is thus left for future work.
>
> [1] Crossformer: Transformer Utilizing Cross-Dimension Dependency for Multivariate Time Series Forecasting. ICLR’23.
>
> [2] iTransformer: Inverted Transformers Are Effective for Time Series Forecasting. ICLR’24.
>
> [3] Unified Training of Universal Time Series Forecasting Transformers. ICML’24.
>
> [4] LightGTS: A Lightweight General Time Series Forecasting Model. ICML’25.
>
> ---
>
> ### **[Q2: Concern about patching rules]**
>
> If the FFT threshold is set overly high or the dominant period is weak or ambiguous, the model is **safeguarded by a fallback mechanism that activates when no dominant frequency can be identified for a given channel.**
>
> As described in Appendix A.2 (”Details of Channel-wise Frequency-based Dynamic Patching”), this fallback mechanism immediately switches to **a sampling-aware patching rule.**
>
> In this regime, we do not enforce a common number of patches across channels; instead, we adjust the relative number of patches according to differences in each channel’s sampling period.
> We clearly observed the benefit of the fallback mechanism in several voyage cases of the private CHS dataset.
>
> Furthermore, even in the more restrictive setting where we ignore channel-wise sampling periods entirely and forcibly unify the patch length across all channels, we do not observe substantial performance degradation on most datasets, as shown in Table 9.

---

> ### Author Response · Authors · 2025-11-20
> **Official Comment by Authors (Cont')**
>
> ---
> ### **[W3: Readability issues in Section 3 & 4]**
>
> We acknowledge the reviewers’ feedback regarding the complexity and clarity of Sections 3 and 4, and we will reorganize these sections in the revised version as follows:
>
> - In Section 3, we will **simplify the problem formulation to more intuitively present the concepts of asynchronicity and test-time block missingness.** Detailed explanations related to the proposed method will be moved to Section 4.
> - We will revise and **disambiguate overlapping notations.**
> - In Section 4, we will describe the dynamic patching strategy and the channel token mechanism **in an algorithmic format.** Based on this, we will provide a more detailed explanation of the implementation and operational procedure of our proposed methodology to improve readability. Due to space limitations, the algorithm itself can be found in our response to reviewer *zsWM*.
> - For the description of the Unified Mask-Guided Attention in Section 4, we will **convert the current inline formulas into numbered equations.**
>
> These modifications will be reflected in the revised version of the paper.
>
> ---
> ### **[W4, Q3: Concern about datasets]**
>
> We note that the benchmarks themselves (e.g., ETT) already involve substantial preprocessing to obtain complete, regularly sampled grids, typically by aggregating higher-frequency sensor streams (such as load-related channels) into 15-minute or hourly averages. This procedure reduces real-world statistical complexity and introduces its own form of synthetic regularity. Our “practical” variants do not create new values, but **instead relax this enforced uniformity by selectively downsampling channels whose variability, assessed in the frequency domain, indicates sufficient oversampling**, so that the resulting multi-rate sensing more plausibly reflects real-world conditions while not materially altering their temporal dynamics.
>
> For missingness injection, we vary the overall missing ratio and inject missingness at the block level by sampling blocks independently for each series and each sample, **mirroring naturally occurring block-wise gaps caused by short-term sensor malfunctions or logging failures in real deployments**. This creates a range of challenging, randomly located missing blocks that are applied identically to all methods.
>
> We also evaluate models on various unmodified real-world datasets, namely the EPA dataset (public) and the CHS dataset (private), where no resampling or missingness injection is applied. **These real-world datasets aggregate diverse cities (EPA) and voyages (CHS)**, and CTF achieves the best performance among the compared methods, which supports its robustness under realistic channel-wise asynchronous sampling and missingness patterns.
>
> ---
> ### **[W5: Clarification on hyperparameter specification]**
>
> It appears that the wording in Appendix A.2 may have caused some misunderstanding.
>
> - *For Transformer-based models, the model dimension dmodel was searched over {128,256,512}, and the feedforward expansion ratio d_ff / d_model was selected from {1,2,4}.*
> - *In our proposed ChannelTokenFormer (CTF), the number of channel tokens was selected from {1,2,3}, with the optimal value determined separately for each dataset.*
>
> Our intent was not to obscure any details; rather, because the hyperparameters vary across datasets, we listed the candidate ranges that were used across all experiments in a unified way.
>
> **For reproducibility, we emphasize that the accompanying code explicitly specifies the exact hyperparameter settings for each dataset.**

---

> ### Author Response · Authors · 2025-11-20
> **Official Comment by Authors (Cont')**
>
> ---
> ### **[W6: Suggestion for a mathematical analysis of interpolation-induced spectral effects]**
>
> Our intention in Appendix B is not to introduce new signal-processing theory, but rather to draw on well-established background and results in digital signal processing to explain that linear interpolation acts as an undesirable low-pass filter, introducing spectral distortion at the input level.
>
> A mathematical analysis of how such input-level interpolation distortions propagate all the way to prediction accuracy in a deep Transformer-based architecture would, in our view, require strong and highly idealized assumptions on both the model and the data, and would effectively amount to a full theoretical dissection of a complex forecasting model. Providing such an analysis in a convincing way is beyond the scope of this work.
>
> **Instead, we complement this with a frequency-bias analysis of the predicted time series in Section 6.3.**
>
> Table 6 compares our method against a structurally similar baseline that, in our practical setting, relies on interpolation at the input level, and shows that our model exhibits less attenuation of the ground-truth spectral characteristics across frequency bands.
>
> This indirectly supports our claim that avoiding interpolation-induced spectral distortion at the input stage leads to better preservation of frequency characteristics in the forecasts. A formal theoretical treatment of these spectral effects is therefore left for future work.
>
>
> ---
> ### **[W7, Q4: Possible failure modes for CTF]**
>
> There is indeed a separate line of work on extremely irregular and highly sparse time series, and methods such as Hi-Patch and tPatchGNN (used as our baselines) are specifically designed for this irregular setting. However, these methods target short-term forecasting on intrinsically very sparse data, and we empirically found that they are not particularly effective in our practical setting.
>
> In contrast, our model is designed for **long-term multivariate forecasting under channel-wise asynchrony, block-wise missingness, and channel dependency**, which reflect realistic conditions in diverse domains. We do not claim to cover all challenges. In particular, our approach is not tailored to cases with extreme, unstructured irregularity and highly bursty missingness.
>
> However, the reviewer’s examples of non-stationary environments and low cross-channel redundancy fall within our target regime. In practice, our benchmarks already exhibit non-stationarity, and when cross-channel structure is weak, our architecture essentially behaves like a strong per-channel model without being harmed by the channel-token mechanism. Moreover, scenarios in which hundreds of sensors are all fully heterogeneous and operate at completely unrelated sampling periods across channels are, in our experience, extremely rare in real-world applications.
>
> ---
> ### **[W8, Q5: Concern about limited superiority in conventional settings]**
>
> We respectfully disagree with the implication that Table 15 suggests limited practical effectiveness of our model.
>
> Our primary goal is **not to set a new state of the art in the idealized case of fully observed, regularly sampled data.**
>
> Instead, the model is designed for realistic multivariate settings with channel-wise asynchrony and block-wise missingness, with the aim of handling these challenges without incurring performance degradation.
>
> Table 15 is provided only to show that, even in the conventional fully observed setting, **our model remains competitive with existing methods and is not over-specialized to our proposed challenging setting.**

---

### Author Response · Authors · 2025-12-02
**Summary for Area Chair**

Dear Area Chair,

We sincerely appreciate you taking the time to evaluate our submission under these exceptional circumstances.
To support your assessment, we provide a concise and factual overview of what occurred during the discussion phase.

We would like to emphasize that **we have strictly adhered to ICLR's double-blind policy throughout the entire review and discussion process.**

During the discussion period, before the emergency stop and rollback to the initial reviews, we engaged in exchanges with two of the four reviewers (9xPZ and zsWm).

- Both had initially assigned relatively low scores based on concerns that were later clarified, but after our rebuttal they reassessed the manuscript positively and raised their ratings to an acceptance level, without changing their confidence scores, increasing the mean score from 4.0 to 5.5.

**These changes in score reflected our detailed rebuttal and the reviewers' careful reconsideration of the paper.**
Although the rollback has reset the scores to their initial values, we fully understand and respect this policy decision.

|Reviewer|Rating|Summary of Review & Discussion|
|---------|---------------|---------------------------------|
|657H|4 (No discussion)|Acknowledged **the strengths of our problem formulation, the corresponding modeling choices, and our extensive empirical validation**. Raised eight concerns and five questions (including questions on *theoretical grounding* and *practical aspects of dataset construction*). We **clarified misunderstandings, highlighted parts of the paper where relevant material was already present, and provided detailed responses to all concerns and questions, revising the manuscript accordingly**. However, no further response was received from the reviewer during the discussion phase.|
|9xPZ|2 → 6|Initially gave a low score based on concerns about *our handling of discrete missing values* and *a lack of comparison with recent channel-dependent baselines*. These concerns largely stemmed from misunderstandings of the existing experimental setup and baseline coverage. We **clarified and addressed all concerns and questions, added additional experimental results and explanations, and revised the manuscript accordingly**. After these updates, **the reviewer commented that the changes substantially improved the overall quality and clarity of the paper, and raised the score to an acceptance level.**|
|e4Zq|6 (No discussion)|Highly recognized **the strengths of our problem formulation and the corresponding unified modeling**. Also noted that the paper is **exceptionally clear, well written, and well illustrated, especially with the detailed appendix**. We responded carefully to the additional questions and requests, including *comparisons with frequency-based methods*, and **incorporated the corresponding clarifications and additional experimental results into the revised manuscript**. However, no further response was received from the reviewer during the discussion phase.|
|zsWm|4 → 6|Initially acknowledged **the strengths of our problem formulation** but found *some of our design choices unclear*. We then provided **a clearer explanation of our unified modeling framework, including algorithmic descriptions for frequency-based dynamic patching and tokenization**. Also clarified our contributions by highlighting the key components and explicitly comparing them to prior work. **After revising the paper to further enhance readability, the reviewer also raised the score to an acceptance level**.|

We hope this summary provides useful context for your recommendation.

For a more detailed report covering our core contributions and a summary of the reviewers’ feedback together with our corresponding comments and revisions, please refer to our next comments, *“Comprehensive Report”*.

---

> ### Author Response · Authors · 2025-12-02
> **Comprehensive Report (1/2)**
>
> ## Our Core Contributions
>
> Our work tackles three core challenges in real-world multivariate time series forecasting simultaneously:
>
> - **Channel-wise asynchronous sampling (A)** arising from heterogeneous sensors in sensor-fusion settings
> - **Long, continuous missing blocks at test time (B)** beyond scattered or sporadic missingness
> - **Noisy and complex inter-channel dependencies (C)**
>
> While prior studies have typically addressed these issues in isolation, to the best of our knowledge no existing approach, including methods for irregular settings, robustly handles all three together.
>
> We further highlight that, under channel-wise asynchrony and block-wise missingness, **interpolation or imputation inevitably introduces input-level signal distortion**, which can degrade forecasting performance regardless of downstream architectural improvements.
>
> To mitigate this, we propose an **interpolation-free framework that integrates targeted strategies for A, B, and C into a unified architecture** explicitly designed to accommodate their complex interplay.
>
> Empirically, we validate this framework on both *modified multivariate benchmarks* with carefully induced asynchrony and block-wise missingness, and *real-world datasets* that naturally exhibit these complex phenomena.

---

> ### Author Response · Authors · 2025-12-02
> **Comprehensive Report (2/2)**
>
> ## Summary of the Reviewers’ Feedback and Our Corresponding Comments & Revisions
>
> *Reviewer 657H*
>
> - **Suggestions for Theoretical Groundings**
>   - A mathematical treatment of robustness in our setting, including interpolation-induced spectral effects on forecasting, *would require highly idealized assumptions and is beyond the scope of this work.*
> - **Concern about Patching Rules (Appendix A.2)**
>   - When no dominant frequency is detected, our CTF automatically falls back to *a sampling-aware patching rule* that adjusts the number of patches per channel based on its sampling period.
> - **Validity of Modified Multivariate Benchmarks (Appendix A.1)**
>   - Provide additional justification for our dataset configurations, explaining that standard benchmarks already involve heavy preprocessing (e.g., aggregation to fixed grids).
>   - *Relax enforced uniformity via frequency-guided downsampling of oversampled channels*, yielding more realistic multi-rate sensing patterns without materially altering the underlying temporal dynamics.
> - **Readability Issues (Sections 3 & 4, Appendix A.2)**
>   - Simplify the problem definition (Section 3)
>   - Enhance the explanation with *algorithmic descriptions* (Section 4, Appendix A.2)
> - **Possible Failure Modes and Target Regime**
>   - Clarify that CTF is primarily designed for *long-term multivariate forecasting under three core challenges*, rather than extremely sparse, highly irregular series targeted by methods such as Hi-Patch.
>   - Also note that our benchmarks *already exhibit non-stationarity*, and that *in low cross-channel redundancy regimes* our architecture behaves essentially like a strong per-channel model.
> - **Concern about Limited Superiority in Conventional Settings (Appendix D)**
>   - Not to establish a new state of the art on fully observed, regularly sampled data, but to handle realistic multivariate conditions without degrading performance.
>   - Even in conventional settings, CTF *remains competitive and is not over-specialized to the challenging conditions we propose*.
> - **Clarification on Hyperparameter Specification (Appendix A.2)**
>   - State that detailed hyperparameter settings for each dataset are fully available in the code provided in the supplementary material.
>
> ---
> *Reviewer 9xPZ*
>
> - **Clarification of Problem Scope and Focus (Introduction)**
>   - Clearly motivate *why our study focuses on three core challenges*, emphasizing that they are particularly impactful in practice yet often oversimplified in conventional multivariate settings.
> - **Concern about Handling Discrete Missingness (Introduction, Section 5, Appendix C.6)**
>   - Point out that discrete missing values are *already present in standard benchmarks* (e.g., ETTm).
>   - Appendix C.6 provides *a detailed analysis, combining embedding visualizations, theoretical reasoning (smooth neural mappings), and additional experiments*, to demonstrate that the small patch embedder exhibits inherent robustness to scattered missing values.
> - **Expanded Related Work & Experiments on Recent Channel-Dependent Approaches (Sections 2 & 5)**
>   - Update the related work on channel-dependent strategies to *include recent advances that involve sparse/clustered inter-channel structures*.
>   - Add experimental comparisons with the latest 2025 channel-dependent baselines (DUET, TimeFilter).
>   - *CTF continues to achieve the best or near-best performance across benchmarks*.
> - **Clarification on Computational Complexity and Limitation (Section 4, Conclusion, Appendix C.3)**
>   - Acknowledge quadratic complexity in the total number of tokens due to unified attention.
>   - But show that *inference latency remains comparable* to strong baselines for realistic channel counts (200 channels).
>   - Note that ultra–high-dimensional regimes (thousands of channels) require additional mechanisms (e.g., channel sparsification), which we identify as important future work.
> - **Additional Details (Appendix A.2)**
>   - Detailed hyperparameter settings - *same as for Reviewer 657H*
>   - Clarify that we *do not use “drop-last” trick*.
> - **Readability Issues** - *same as for Reviewer 657H*
>
> ---
> *Reviewer e4Zq*
>
> - **Comparison with Non-Transformer / Frequency-based Baselines (Appendix C.5)**
>   - Already compared against strong non-Transformer models (TimeMixer++, CrossGNN, etc.), and *added further baselines with frequency-based methods (FreTS, FITS)*.
> - **Frequency Bias & Implications for Forecasting Performance (Appendix B.3)**
>   - Show that interpolation-induced input distortions cause frequency bias and degrade forecasting performance, motivating interpolation-free architectures.
>   - *Supported by additional experiments on frequency-based baselines and a frequency-bias analysis of the predicted series*.
> - **Limitation in Computational Complexity** - *same as for Reviewer 9xPZ*
>
> ---
> *Reviewer zsWm*
>
> - **Presentation of Design Choices and Contributions (same as readability issues)**  - *same as for Reviewer 657H, 9xPZ*

---

### Meta-Review · Area_Chair_3ZDZ · 2026-01-04

**Summary:**

The authors propose an interpolation-free framework that integrates targeted strategies into a unified architecture explicitly designed to accommodate their complex interplay. Most reviews are positive. Specifically, 657H acknowledged the strengths of the problem formulation, the corresponding modeling choices, and  extensive empirical validation. 9xPZ commented that the changes substantially improved the overall quality and clarity of the paper, and raised the score to an acceptance level.
e4Zq highly recognized the strengths of our problem formulation and the corresponding unified modeling. Also noted that the paper is exceptionally clear, well written. zsWm	acknowledged the strengths of the problem formulation but found some of our design choices unclear. After revising the paper to further enhance readability, the reviewer also raised the score to an acceptance level. The authors' rebuttal has adeptly addressed the majority of concerns raised by the reviewers.
Overall, I recommend the acceptance of this submission.

**Reviewer Concerns:**

Most concerns of 9xPZ and zsWm have been solved, since  9xPZ raised the score from to 2 to 6, and zsWm	raised the score from 4 to 6. For others, part of concerns have not been solved.

**Reviewer Scores:**

9xPZ and zsWm would raise the score.

---

### Decision · Program_Chairs · 2026-01-26

Accept (Poster)